# Incomplete removal of extracellular glutamate controls synaptic transmission and integration at a cerebellar synapse

Timothy S Balmer[1†], Carolina Borges-Merjane[1,2‡], Laurence O Trussell[1*]

[1]Vollum Institute and Oregon Hearing Research Center, Oregon Health & Science University, Portland, United States; [2]Neuroscience Graduate Program, Vollum Institute, Oregon Health & Science University, Portland, United States

**Abstract** Synapses of glutamatergic mossy fibers (MFs) onto cerebellar unipolar brush cells (UBCs) generate slow excitatory (ON) or inhibitory (OFF) postsynaptic responses dependent on the complement of glutamate receptors expressed on the UBC's large dendritic brush. Using mouse brain slice recording and computational modeling of synaptic transmission, we found that substantial glutamate is maintained in the UBC synaptic cleft, sufficient to modify spontaneous firing in OFF UBCs and tonically desensitize AMPARs of ON UBCs. The source of this ambient glutamate was spontaneous, spike-independent exocytosis from the MF terminal, and its level was dependent on activity of glutamate transporters EAAT1–2. Increasing levels of ambient glutamate shifted the polarity of evoked synaptic responses in ON UBCs and altered the phase of responses to in vivo-like synaptic activity. Unlike classical fast synapses, receptors at the UBC synapse are virtually always exposed to a significant level of glutamate, which varies in a graded manner during transmission.

*For correspondence:
trussell@ohsu.edu

Present address: †School of Life Sciences, Arizona State University, Tempe, United States; ‡Cellular Neuroscience, Institute of Science and Technology Austria, Klosterneuburg, Austria

Competing interests: The authors declare that no competing interests exist.

## Introduction

The neurotransmitter glutamate initiates phasic postsynaptic currents following its release from synaptic vesicles and binding to ionotropic postsynaptic receptors. Glutamate is then cleared within milliseconds from the synaptic cleft by diffusion and the activity of excitatory amino acid transporters (EAATs). Rapid glutamate removal is essential to maintain brief and reliable transmission. By contrast, unipolar brush cells (UBCs) of the cerebellar cortex transform brief mossy fiber (MF) excitation into a prolonged postsynaptic current due to the persistence of glutamate in the synapse for hundreds of milliseconds after synaptic release (*Lu et al., 2017*; *Rossi et al., 1995*). The unusually long lifetime of glutamate at this synapse may be due to a barrier to diffusion caused by the convoluted dendritic brush (*Rossi et al., 1995*), perhaps combined with a relatively low expression of EAATs. Due to these constraints, removal of glutamate is not only slow, but might also be incomplete. If so, the mechanism of synaptic transmission would be quite unique at this synapse, with transmission being mediated by elevations in free glutamate above a significant baseline level, while always maintaining a near equilibrium of transmitter with receptors.

At some synapses, ambient neurotransmitter can induce tonic currents (also called background or standing currents). For example, cerebellar granule cells have a tonic inhibitory current due to ambient GABA-activating extrasynaptic $GABA_A$ receptors (*Brickley et al., 1996*; *Kaneda et al., 1995*). At other synapses, tonic activation of high-affinity NMDARs (*Sah et al., 1989*) and metabotropic receptors *Bandrowski et al., 2003* have been reported. We investigated whether ambient glutamate and tonic currents are a prominent feature of the MF to UBC synapse, given the unusual time course of glutamate signaling that characterizes this synapse.

This question was of particular interest because tonic glutamate receptor currents could have profound effects on cerebellar activity. UBCs are spontaneously active in vivo (*Ruigrok et al., 2011*; *Simpson et al., 2005*), and their axons terminate in multiple MFs that likely coordinate the firing of numerous granule cells and their parallel fibers. Thus, in addition to more typical intrinsic mechanisms (*Diana et al., 2007*; *Russo et al., 2007*), tonic glutamate currents at a single UBC could control the UBCs' spontaneous firing rate and therefore the activity of downstream neurons.

Here we report that a physiologically significant level of ambient glutamate is indeed present at the MF to UBC synapse, where it controls spontaneous firing by tonic mGluR2 activation and regulates phasic postsynaptic currents by partially desensitizing AMPA receptors (AMPARs). The source of the ambient glutamate is not action potential-evoked release, but is due to spontaneous vesicle fusion, and its level is regulated by EAATs. Indeed, stimulating the MF to UBC synapse with in vivo-like frequency patterns shows that EAATs control the phase delay of firing between MF input and UBC output that likely coordinates the activity of hundreds of parallel fibers and may play a role in cerebellar learning.

## Results

### Ambient glutamate is regulated by EAATs

EAATs play an important role in synaptic transmission by removing glutamate from the synapse after vesicular release. At the MF to granule cell synapse, EAATs limit the exposure to glutamate spillover (*DiGregorio et al., 2002*; *Overstreet et al., 1999*; *Xu-Friedman and Regehr, 2003*). At the same cerebellar glomeruli, MFs release glutamate onto ON UBCs and OFF UBCs, which, respectively, give a net inward and outward current response (*Borges-Merjane and Trussell, 2015*), and EAATs limit the duration of excitatory postsynaptic currents (EPSCs) and reduce AMPAR desensitization (*Lu et al., 2017*). The outward current results from mGluR2-activated $K^+$ conductance while inward current results from AMPAR. We tested the hypothesis that EAATs control *ambient* glutamate levels at the MF to UBC synapse by making whole-cell recordings from UBCs in lobe X (nodulus) of mouse cerebellum. ON or OFF UBC subtypes were differentiated by their response to pressure ejection of glutamate (1 mM, puff) via a second pipette or electrical stimulation of presynaptic MFs. In all experiments, $GABA_A$, glycine and mGluR1 receptors were blocked (see Materials and methods). In these and some previous experiments (*Borges-Merjane and Trussell, 2015*), we did not observe synaptic activation of mGluR1, despite the sensitivity of the cells to agonists of these receptors.

In the first set of experiments, we reasoned that blockade of EAATs should elevate ambient glutamate, leading to tonic inward or outward currents, depending on the UBC subtype and its complement of receptors. In OFF UBCs, bath application of EAAT inhibitor DL-TBOA (50 µM) caused an outward shift in holding current that was blocked by mGluR2 antagonist LY341495 (1 µM) (*Figure 1A,B*), consistent with EAATs limiting ambient glutamate and thus tonic mGluR2 receptor activation. Antagonist application made the holding current inward relative to control, presumably because of a tonic AMPAR-mediated current. In addition to an outward shift in holding current, the mGluR2-mediated outward currents became smaller in the presence of DL-TBOA, suggesting that ambient glutamate activated synaptic receptors (*Figure 1C*).

To directly test for tonic AMPAR current, ON UBC responses were recorded in LY341495. DL-TBOA now caused an inward shift in holding current, blocked by the AMPAR antagonist GYKI53655 (GYKI, 50 µM) (*Figure 1D–F*). In addition to tonic inward AMPAR current, increased ambient glutamate caused three other effects, as previously described (*Lu et al., 2017*): (1) the steady-state inward current during synaptic stimulation decreased and in many cases shifted outward, as in the example in *Figure 1F*, (2) the slow EPSC, after stimulation lengthened in duration (*Figure 1Fi*), and (3) the fast (phasic) EPSC amplitudes were reduced (*Figure 1Fii*). Therefore, EAATs not only limit glutamate accumulation at rest and hasten its removal after release, but shape the time course and amplitudes of synaptic currents.

To address which subtypes of EAATs contribute to glutamate uptake at this synapse we pharmacologically isolated AMPAR current and utilized the differing affinities of EAAT1-3 for antagonists DL-TBOA and TFB-TBOA (*Figure 1—figure supplement 1*; *Shimamoto et al., 1998*; *Shimamoto et al., 2004*). A low concentration of TFB-TBOA, which blocks EAAT1-2, generated a tonic inward current (*Figure 1—figure supplement 1A–C*). Increasing the concentration to

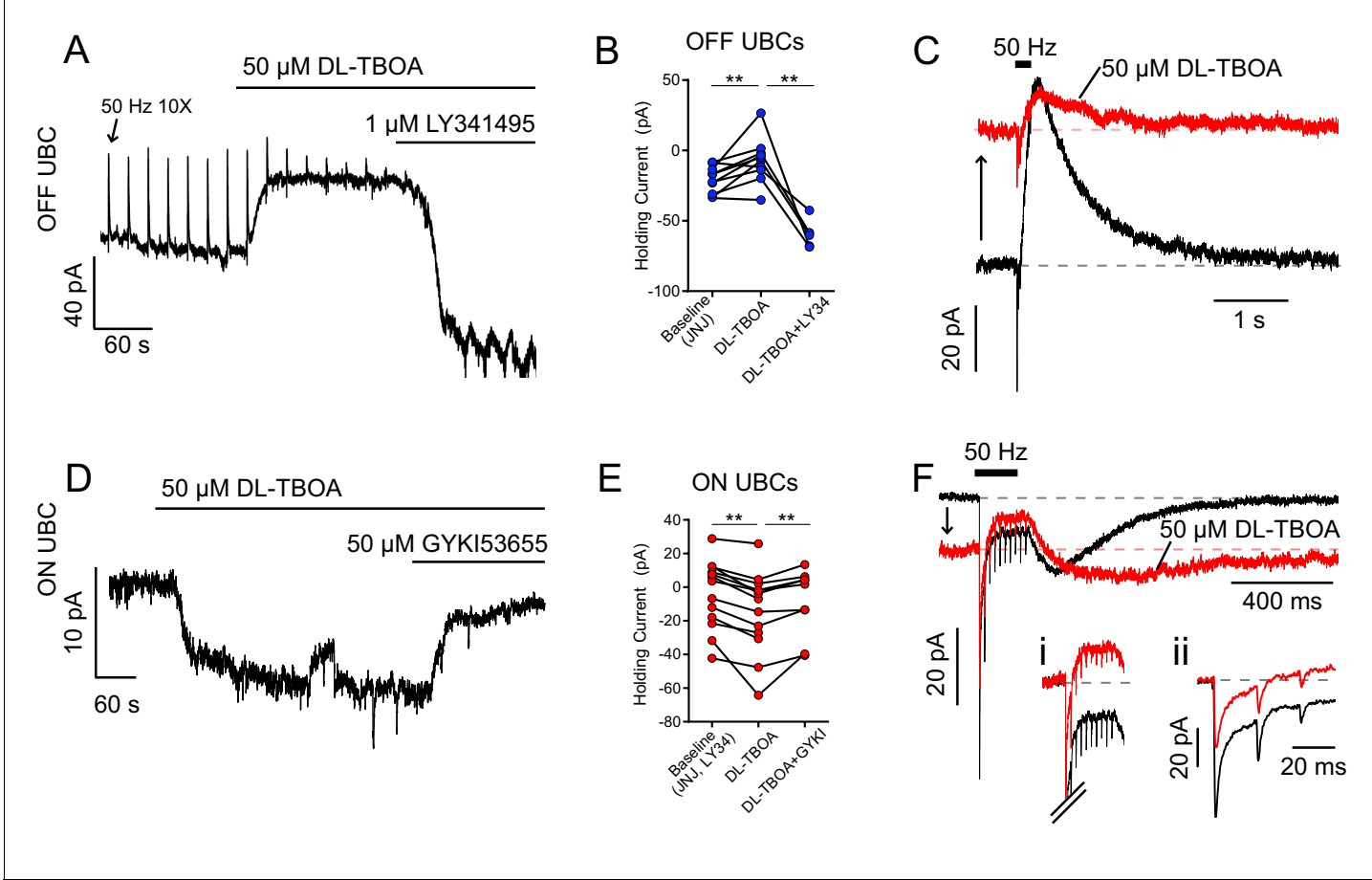

**Figure 1.** EAATs prevent glutamate accumulation at rest. (**A**) OFF UBC responding to 50 Hz 10 × 10 V electrical stimulation of white matter every 20 s. The upward deflections are mGluR2-mediated outward currents typical of OFF UBC responses to synaptic stimulation (arrow points to the first). DL-TBOA caused outward shift in tonic current, blocked by mGluR2 antagonist LY341495. (**B**) Effect of DL-TBOA on OFF UBCs. Paired t-tests, baseline vs DL-TBOA, p=0.008, n = 10; DL-TBOA vs DL-TBOA+LY34, p=0.006, n = 5. (**C**) Synaptic currents evoked by electrical stimulation. Same cell as in (**A**). The holding current shifted outward (upward arrow) when EAATs were blocked by DL-TBOA. (**D**) ON UBC example. DL-TBOA caused an increase in tonic inward current, blocked by AMPAR antagonist GYKI. (**E**) Effect of DL-TBOA on ON UBCs. Paired t-tests, baseline vs DL-TBOA, p=0.0003, n = 13; DL-TBOA vs DL-TBOA+GYKI, p=0.007, n = 9. (**F**) ON UBC synaptic currents evoked by 50 Hz 10× electrical stimulation. DL-TBOA caused the holding current to shift inward (downward arrow) and the slow rebound EPSC became slower to rise and decay. (**Fi**) Steady-state current during stimulation, with the tonic current subtracted to show that in DL-TBOA the steady-state current is outward relative to the tonic current. Scale as in (**F**). (**Fii**) Phasic EPSCs with tonic currents subtracted. The first EPSC became smaller due to increased tonic AMPAR desensitization. The holding potential was maintained at −70 mV, and thus increasing AMPAR currents are inward and increasing mGluR2 responses are outward.

The online version of this article includes the following source data and figure supplement(s) for figure 1:

**Figure supplement 1.** EAAT1–2 contribute to glutamate uptake at MF-UBC synapse.

**Figure supplement 2.** AMPAR kinetic model predicts glutamate transients that produce slow EPSC.

**Figure supplement 2—source data 1.** The parameters of the AMPA receptor kinetic model were adjusted to give incomplete desensitization and to fit experimental data from *Lu et al., 2017*.

**Figure supplement 3.** Some AMPAR currents recorded in DL-TBOA required increasing the maximal conductance to produce a good fit, suggesting that DL-TBOA allows spillover onto additional receptors not activated when glutamate transporters are functioning.

additionally block EAAT3 had no effect. The inward current induced by the drug was caused by an increase in ambient glutamate that activated AMPARs, as it was blocked by GYKI. By contrast, a low concentration of DL-TBOA, which blocks EAAT2-3, and a high concentration that additionally blocks EAAT1, both shifted the holding current inward (*Figure 1—figure supplement 1D,E*). Thus, both EAAT1 and EAAT2 contributed to glutamate uptake at this synapse.

## Prediction of ambient and evoked glutamate concentration profiles

To explore how EAATs contribute to the generation of ultra-slow signaling through AMPARs at the UBC synapse, a 13-state kinetic model was constructed that had five closed states representing occupation of 0–4 binding sites (C0–C4) and transitions from these closed-bound states to four open (O1–O4) and four desensitized states (D1–D4) (*Figure 1—figure supplement 2A*). Rates were adjusted to give incomplete desensitization, such that receptors with three or four glutamates bound had lower equilibrium current than receptors with one or two glutamates bound (*Figure 1—figure supplement 2—source data 1*). The model generated currents (*Figure 2B*) that fit the experimental data from *Lu et al., 2017* (dots on *Figure 1—figure supplement 2C*), reproducing the essential features of the steady-state dose–response curves (*Figure 1—figure supplement 2C*).

The model was then used to predict the glutamate concentration at synaptic AMPARs receptors at rest and during synaptic transmission. Trains of glutamate transients were generated using a 3D diffusion equation (see Materials and methods; *Balmer and Trussell, 2021*) and applied to the model AMPARs. This equation was simpler to implement than sums of exponential decays; both approaches could account well for an initial fast decay of transmitter followed by a very slow decay expected from large synaptic spaces. The resulting current was compared to data recorded from UBCs, and parameters of the diffusion equation were then varied until an excellent fit between model and data was obtained for both fast and slow components of the synaptic response (*Figure 1—figure supplement 2D*). This approach revealed that the slow EPSC at the end of the train increased as the glutamate concentration at the receptors fell from >1 mM to the peak of the steady-state dose–response curve (32 µM), and then decayed slowly over hundreds of milliseconds as the concentration gradually decreased below 32 µM (*Figure 1—figure supplement 2E*). Importantly, the glutamate transients could not generate currents that fit the data recorded in the absence of EAAT blockers unless we incorporated a low level of ambient glutamate into the model (*Figure 1—figure supplement 2F*). The error values of the fits (square norm between the fit and the data) were reduced 44.5 ± 24.0% (mean ± SD) by the addition of the ambient glutamate parameter and produced currents that fit the data significantly better (paired t-test, p<0.001). Simulations lacking ambient glutamate consistently showed relatively small slow currents as compared to the experimental records, despite matching the dose–response for exogenous application of glutamate. The model predicts that in order for the characteristic slow EPSC to be produced, not only must glutamate persist at the synapse for hundreds of milliseconds after synaptic release, but a background level of ~5 µM glutamate must be present prior to synaptic stimulation (*Figure 1—figure supplement 2G*).

Having established a glutamate time course that accounts for control data, we then explored how the time course was changed by DL-TBOA. Synaptic responses were recorded in nine UBCs, before and after application of DL-TBOA. For all data, the model was first fit to control data to establish the glutamate transients predicted with EAATs active. Then, responses in DL-TBOA from these same cells were fit by varying only two parameters, the ambient glutamate concentration and the amount of glutamate released. It should be emphasized that the number of glutamate molecules 'released' refers to the number available for diffusion and binding to receptors; EAATs capable of rapidly buffering glutamate will reduce the effective amount of release, as previously described (*Diamond and Jahr, 1997*). For five of the cells, simply increasing ambient glutamate concentration and glutamate released generated simulated AMPAR currents that reproduced the major effects of DL-TBOA in our experiments, including the outward shift of the steady-state current during the stimulation train and a longer duration slow EPSC at the cessation of stimulation (*Figure 1—figure supplement 2H, I*). For these cases, ambient glutamate was increased 2.4-fold, from 4.7 ± 2.7 µM to 11.2 ± 7.6 µM (mean ± SD) and glutamate molecules released were increased 2.3-fold. In three other cells, increasing maximum conductance (i.e. number of receptors) 1.7 ± 0.5 fold (mean ± SD) was also required to modify the glutamate transient to produce a current that fit the DL-TBOA condition (*Figure 1—figure supplement 3*). This suggests that in some cases, glutamate released does not, under normal conditions, reach all receptors on the cell and that when EAATs are blocked, spillover to additional receptors may occur.

The model thus predicts that EAATs do not fully remove transmitter from the synapse at rest, but instead reduce ambient transmitter to a low micromolar level. Based on the estimate of 5 µM ambient glutamate, only 54.2% of AMPAR are free of agonist prior to spike-driven release from the MF

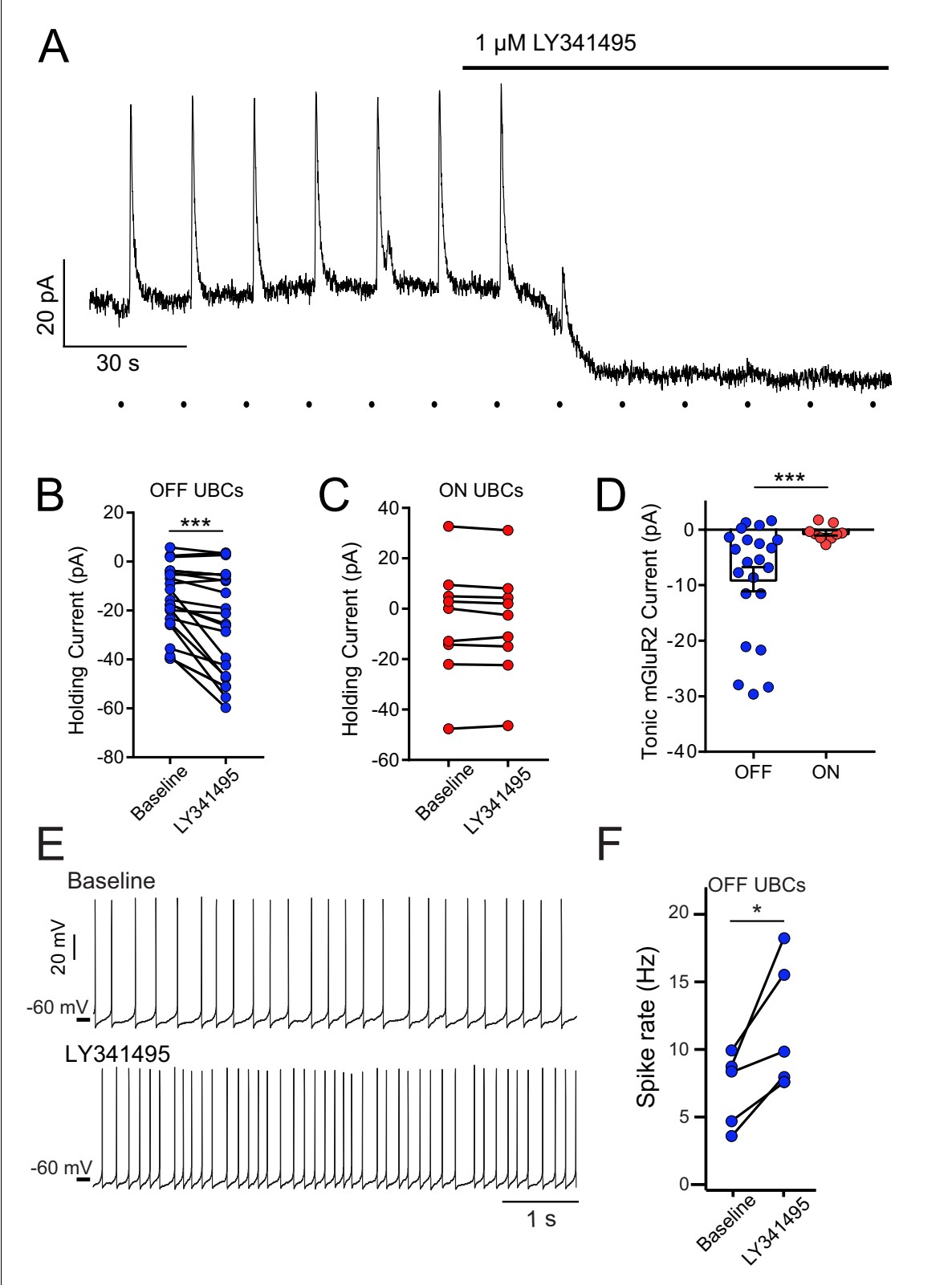

**Figure 2.** mGluR2-mediated tonic outward current in OFF UBCs, but not ON UBCs. (**A**) Example bath application of mGluR2 antagonist LY341495 on OFF UBC (in the presence of 5 µM NBQX, 1 µM JNJ16259685). Synaptic stimulation (100 Hz, 20×, 10 V, 15 s intervals, indicated by dots below trace) evokes an outward current, blocked by LY341495. A ~20 pA standing outward current is also blocked. (**B**) UBCs identified as OFF UBCs based on synaptic stimulation or glutamate puff had an mGluR2 standing current, blocked by LY341495. Paired t-test, p<0.001, n = 22. (**C**) UBCs identified as ON

*Figure 2 continued on next page*

Figure 2 continued

UBCs based on synaptic stimulation or glutamate puff did not have an mGluR2 standing current. Paired t-test, p=0.243, n = 9. (**D**) Summary of mGluR2 standing currents. The tonic mGluR2 current was larger in OFF UBCs than ON UBCs. Student's t-test, p=0.021, n = 31. (**E**) Example OFF UBCs in current-clamp mode before and after LY341495, showing increase in spike rate after blocking tonic mGluR2-mediated current. (**F**) Summary showing significant increase in spike rate across OFF UBCs. Paired t-test, p=0.025, n = 5.

The online version of this article includes the following figure supplement(s) for figure 2:

**Figure supplement 1.** Widely utilized mGluR1 antagonist LY367385 also acts as mGluR2 agonist.

**Figure supplement 2.** Spontaneous activity and ON and OFF responses in cell-attached recordings.

**Figure supplement 3.** Tonic AMPAR current in ON UBCs, but not OFF UBCs.

terminal, with the remainder bound to at least one agonist molecule. Following exocytosis, EAATs restrict the amount of glutamate that reaches AMPARs. When EAATs are blocked, there is a large increase in ambient glutamate and in glutamate available from exocytosis.

## Ambient glutamate causes a tonic outward current in OFF UBCs

The model predicted that micromolar levels of ambient glutamate persist and shape the synaptic currents in ON UBCs. This persistence of glutamate is surprising given that studies documented only low nanomolar levels at other synapses, due to the efficacy of EAAT activity (*Herman and Jahr, 2007*). We therefore tested the model's prediction by investigating whether ambient glutamate is present at the UBC synapse in the absence of EAAT blockers. For OFF UBCs, where mGluR2 receptors activate outward currents, ambient glutamate would be expected if application of an mGluR2 antagonist shifted the holding current inward. To isolate the mGluR2-mediated current, these experiments were done in the presence of a selective non-competitive antagonist of mGluR1 (JNJ16259685). A commonly used antagonist of mGluR1 (LY367385) was not used, as it was found to additionally act as an agonist of mGluR2 (*Figure 2—figure supplement 1*).

*Figure 2A* shows a trace recorded from a voltage-clamped OFF UBC; outward deflections are inhibitory responses to MF stimulation. As expected, LY341495 not only blocked the synaptically evoked outward currents, but also shifted the holding current inward, consistent with blockade of a tonic outward current mediated by mGluR2 receptors (*Figure 2A*). Only OFF UBCs had such a tonic outward current (*Figure 2B–D*), despite both UBC subtypes' expression of the inhibitory mGluR2 (*Jaarsma et al., 1995*), presumably because ON UBCs have significantly less mGluR2-GIRK-mediated current than OFF UBCs (*Borges-Merjane and Trussell, 2015*). Current clamp was then used to characterize the effects of tonic outward current on firing activity in OFF UBCs. UBCs are spontaneously active in vivo and in vitro, firing spikes without synaptic input (*Kim et al., 2012*; *Ruigrok et al., 2011*), and glutamate released from MF serves to either increase or decrease this spontaneous activity (*Figure 2—figure supplement 2*). In the presence of the mGluR2 antagonist, the frequency of spontaneous firing increased significantly (*Figure 2E,F*). Therefore, ambient glutamate at OFF UBC synapses tonically activates mGluR2 receptors and reduces spontaneous spike rate, presumably lowering transmitter release at UBC axon terminals onto granule cells.

## Small tonic AMPAR currents in ON UBCs caused by ambient glutamate

We next investigated the effect of ambient glutamate on AMPARs. In the experiment with an ON UBC (*Figure 2—figure supplement 3A*), bath application of AMPAR antagonist NBQX blocked the evoked inward current and shifted the baseline current outward, consistent with blockade of a tonic inward current. In OFF UBCs, there was no effect of AMPAR antagonists on baseline current (*Figure 2—figure supplement 3B*). Across the population of ON UBCs, there was a small, but significant effect of the antagonist (*Figure 2—figure supplement 3B,C*). However, the illustrated example was an uncommonly large effect; the tonic AMPAR current was typically just a few picoamperes. This small current may be an underestimate, given that during these relatively long recordings UBCs typically become increasingly leaky, which would oppose the blocking effect of NBQX on a small inward current. However, in current-clamp recordings, the spike rate was not consistently affected by the AMPAR antagonist (*Figure 2—figure supplement 3E,F*), unlike the case for mGluR2 block in OFF UBCs. It is possible that ambient glutamate at the ON UBC synapse is low compared to OFF UBCs,

despite the predictions of the model. Alternatively, however, it may be that micromolar levels of ambient glutamate were present, but did not conduct significant tonic inward current because of partial AMPAR desensitization.

## AMPAR desensitization caused by ambient glutamate

Cyclothiazide (CTZ), an allosteric modulator of AMPARs that inhibits desensitization (*Partin et al., 1993*), was used to test whether ambient glutamate leads to tonic AMPAR desensitization. If true, CTZ application would shift the baseline current inward as receptors transition from the desensitized state to the open state. Indeed, in both ON and OFF UBCs, 100 µM CTZ caused a striking inward shift in baseline current that was blocked by NBQX (*Figure 3A–C*). The current was larger and noisier during CTZ than control and was blocked NBQX (*Figure 4A*), consistent with increased channel activity caused by CTZ relieving AMPARs from their desensitized state. CTZ may have presynaptic effects that could induce glutamate release (*Diamond and Jahr, 1995*). Accordingly, the same experiment was done with cerebellar granule cells, which receive the same MF inputs as UBCs (*Balmer and Trussell, 2019*). Despite sharing MF presynaptic input and having similar AMPAR sub-types (*Kinney et al., 1997*), an effect of CTZ on baseline current was not observed in granule cells (*Figure 3D,E*), and thus the effect of CTZ on UBCs is unlikely to be due to the drug's effect on the MF terminal. The absence of an effect of CTZ on granule cells suggests that ambient glutamate is cell specific, rather than pervasive through the brain slice.

If ambient glutamate desensitized AMPARs, it might also reduce synaptically evoked EPSCs. Indeed, synaptic currents in UBCs were significantly larger after relief of desensitization by CTZ, in some cases strikingly so (*Figure 3F–G*). On average, the amplitude of the first EPSC in the train doubled, and the charge evoked by the train increased ninefold, principally due to increase in current between each EPSC (*Figure 3F*), and the decay constant of the slow EPSC lengthened from 265 ± 72 ms to 416 ± 110 ms (mean ± SD, paired t-test, p=0.002, n = 6). The broadened EPSCs evoked by a brief stimulation indicates that elevated glutamate persists at the synapse for seconds, but does not typically evoke a current because AMPARs return to their tonic desensitized state. The slow decay in CTZ is consistent with the prolonged time course for recovery from AMPAR desensitization following transmitter release at the UBC synapse, assayed with either paired synaptic stimuli or glutamate uncaging (*Lu et al., 2017*), confirming that glutamate lingers in the synaptic cleft for hundreds of milliseconds. Thus, tonic desensitization by ambient glutamate constrains the amplitude and duration of synaptic excitation in UBCs.

## The source of ambient glutamate is not action potential-evoked exocytosis

What is the source of ambient glutamate at the MF–UBC synapse? Spontaneous firing of MFs only rarely occurs in acute slices, so action potential-evoked release of glutamate from the MF input to the UBC can be ruled out as the source of ambient glutamate. To test whether spillover into the UBC brush from other spontaneously active neurons in the slice contributes to the tonic glutamate current, TTX was used to block action potential-evoked transmitter release, with the expectation that this manipulation should then reduce the size of the tonic current. TBOA was first applied to increase the ambient glutamate to enhance sensitivity of the test. These experiments were conducted in the presence of strychnine, SR99531, JNJ16259685, and MK-801 to isolate mGluR2 and AMPAR currents.

In OFF UBCs, application of 1 µM TTX had no effect on the tonic current, and 1 µm LY341495 blocked an ambient mGluR2 current, similar to its effects in controls (*Figure 4A,B*). Thus, the tonic current in OFF UBCs is not dependent of action potential firing. By contrast, in ON UBCs, TTX did block a tonic inward current (*Figure 4C,D*). However, UBCs have a persistent $Na^+$ current that could underlie the effect of TTX (*Afshari et al., 2004*; *Russo et al., 2007*). GYKI53655 (50 µM) was initially applied to block AMPARs prior to applying TTX in order to test whether TTX had a direct effect on a $Na^+$ channel conducting a tonic inward current. We found that application of TTX blocked an inward current whose amplitude was the same whether or not GYKI was present (*Figure 4E,F*), indicating that the sole effect of TTX was to block a persistent $Na^+$ current. Moreover, TTX had little effect on glutamate puff-evoked mGluR2 currents in OFF UBCs (*Figure 4G*) or AMPAR currents in ON UBCs (*Figure 4H*). Taken together, these results indicate that ambient glutamate is specific to

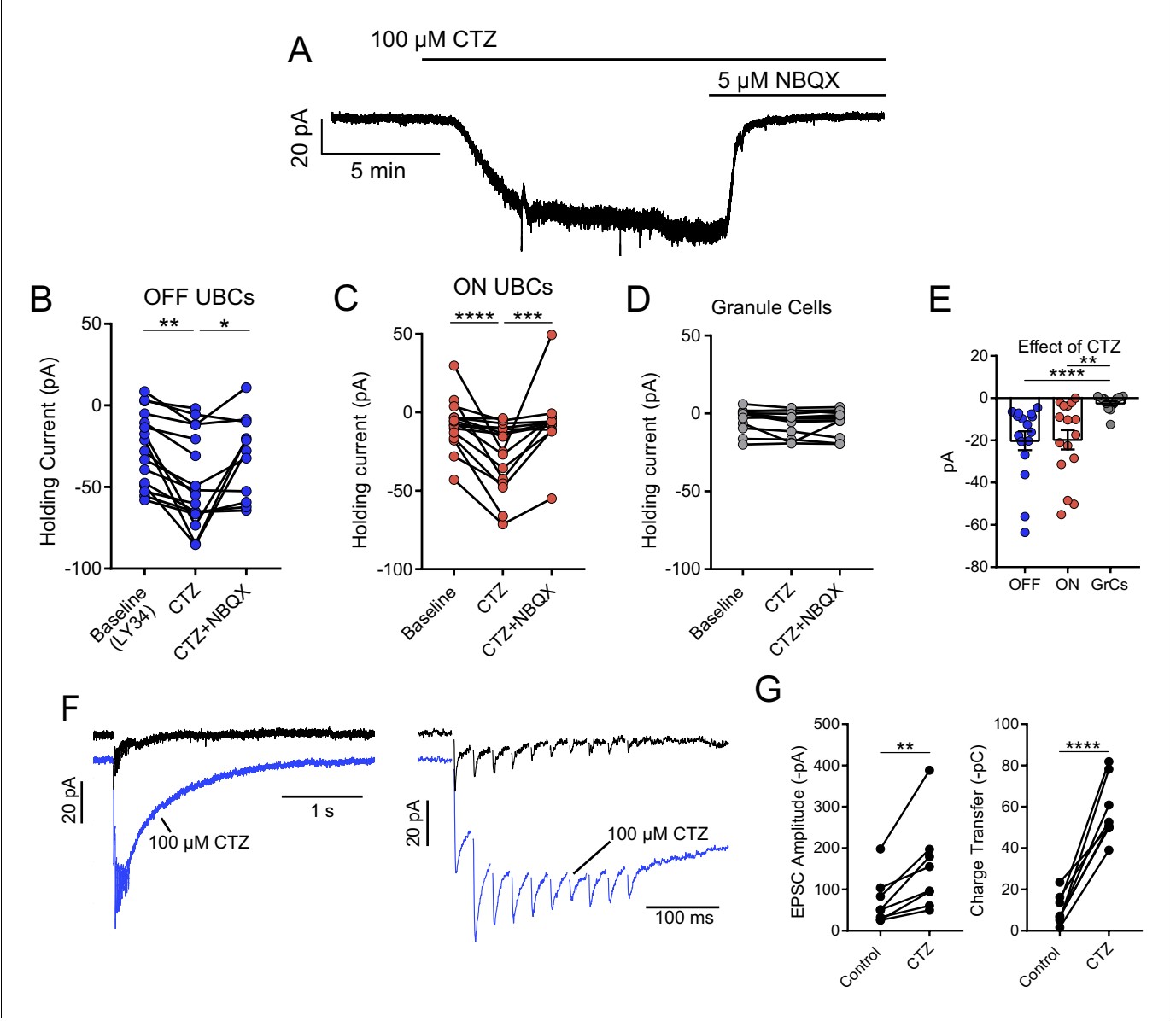

**Figure 3.** Tonic AMPAR desensitization in UBCs, but not granule cells. (**A**) 100 µM cyclothiazide (CTZ) bath application increases standing inward current of this OFF UBC and is blocked by 5 µM NBQX. Note increase in noise in CTZ, presumably due to increased AMPAR activity. (**B**) CTZ increased inward current in OFF UBCs, indicating ambient level of AMPAR desensitization. Wilcoxon signed rank: control vs CTZ, p=0.014, n = 16; paired t-test: CTZ vs CTZ+NBQX, p=0.025, n = 11. (**C**) CTZ increased inward current in ON UBCs. Paired t-tests: control vs CTZ, p=0.0001, n = 16; CTZ vs CTZ +NBQX, p=0.001, n = 12. (**D**) There was no significant effect of CTZ on the holding current of granule cells: Friedman test: p=0.264, n = 12. (**E**) The effect of CTZ on OFF and ON UBCs was not different, but both were significantly different than on GrCs. Mann–Whitney test: OFF UBCs vs GrCs, p<0.0001, n = 29; unpaired t-tests: ON UBCs vs GrCs, p=0.001, n = 29. OFF UBCs vs ON UBCs, p=0.809, n = 32. (**F**) ON UBC synaptic responses to 50 Hz, 10× electrical stimulation and response after CTZ. Right: EPSCs evoked during 50 Hz stimulation, magnified. The first EPSC became much larger, indicating that ambient glutamate desensitizes AMPARs. (**G**) CTZ increased the EPSC amplitude 2.3 ± 0.8 fold (mean ± SD), paired t-test, p=0.006, n = 8. The charge during the 50 Hz, 10× stimulation, and the long inward tail increased 9.3 ± 7.0 fold (mean ± SD). Paired t-test, p<0.0001, n = 8.

the UBC and not to neighboring granule cells and is independent of action potential firing in the slice. A likely source for this glutamate is therefore from spontaneous exocytic events.

## Spontaneous exocytosis is a source of ambient glutamate
Spike-independent, spontaneous exocytosis of glutamate generates miniature excitatory synaptic currents (mEPSCs) in most central neurons. UBCs were recorded in the presence of TTX in order to

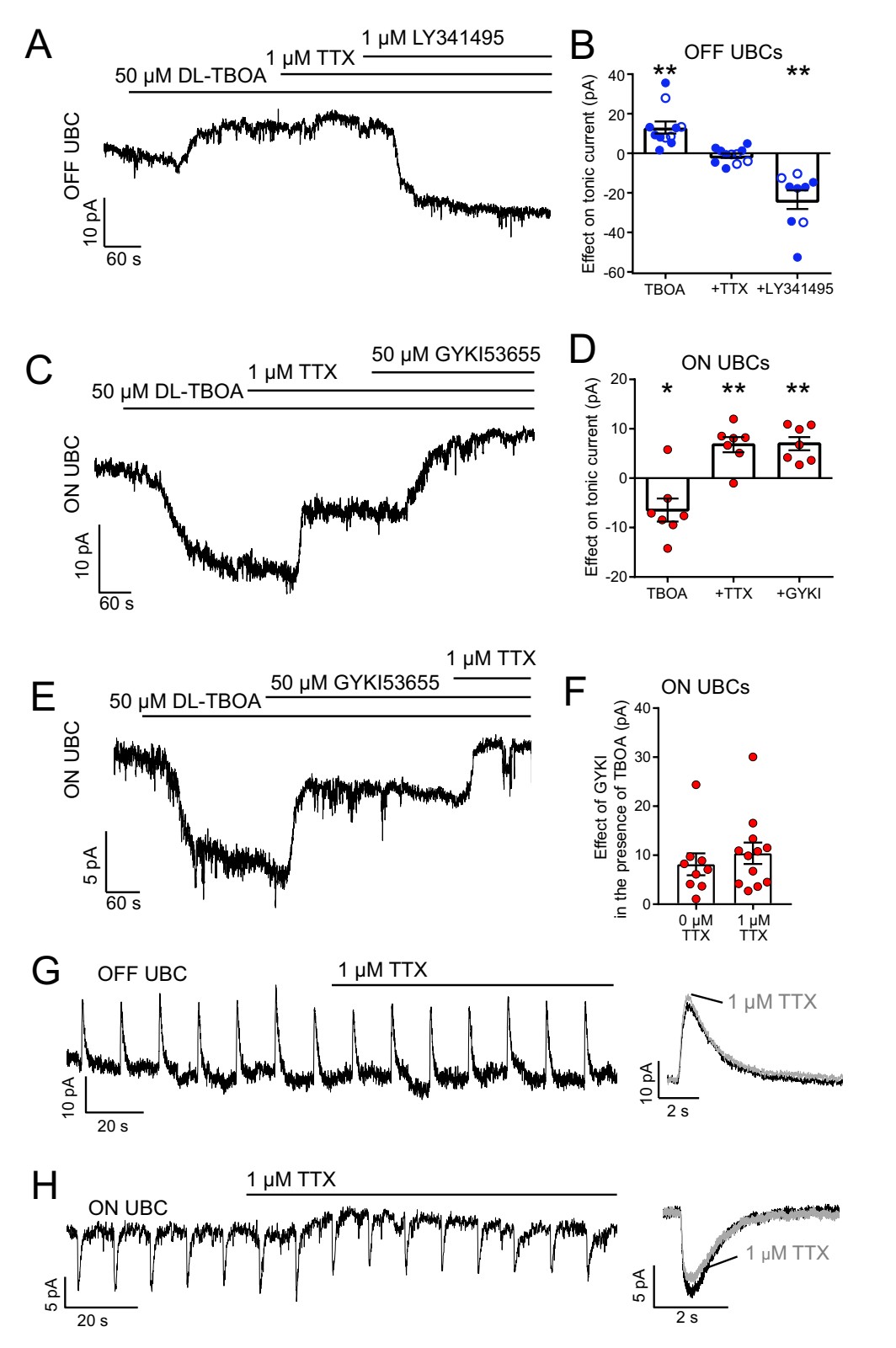

**Figure 4.** The source of ambient glutamate is not action potential-evoked release. (**A**) Example tonic currents in OFF UBC. DL-TBOA increased tonic outward current in all OFF UBCs. Addition of TTX did not reverse this effect. Addition of mGluR2 antagonist LY341495 (1 μM) blocked the outward current, shifting the tonic current inward. (**B**) Summary of the effect of each additional agent in seven OFF UBCs. On average, there was no effect of TTX on the tonic outward current of OFF UBCs. In some cases, GYKI53655 (50 μM) was applied before DL-TBOA (unfilled circles), but this had no effect

*Figure 4 continued on next page*

*Figure 4 continued*

and were therefore pooled with examples in the absence of GYKI53655 (filled circles). Paired t-tests: Baseline vs DL-TBOA, p=0.0011, n = 11; DL-TBOA vs DL-TBOA+TTX, p=0.674, n = 11; DL-TBOA+TTX vs DL-TBOA+TTX+LY341495, p=0.0013, n = 9. (**C**) Example tonic currents in ON UBC. After determination of ON or OFF UBC type by synaptic stimulation or glutamate puff, LY341495 was added before DL-TBOA. DL-TBOA increased tonic inward current. TTX reduced inward current. GYKI further reduced the inward current. (**D**) Summary of the effect of agents in seven ON UBCs. Paired t-tests: baseline vs DL-TBOA, p=0.033, n = 7; DL-TBOA vs DL-TBOA+TTX, p=0.004, n = 7; DL-TBOA+TTX vs DL-TBOA+TTX+GYKI53655, p=0.002. (**E**) To test whether TTX was blocking a persistent sodium current that is known to be present in UBCs, GYKI was applied first. Any effect of TTX after GYKI was interpreted as block of persistent sodium current. This example ON UBC had a persistent sodium current that was blocked. (**F**) Summary of the effect of GYKI in the presence or absence of TTX. GYKI blocked the same amount of tonic AMPA current whether or not action potentials were blocked by TTX, indicating that the sole effect of TTX was to block a persistent $Na^+$ current. Unpaired t-test: p=0.486, n = 21. (**G**) TTX had little effect on the response of OFF UBCs to 10 ms 1 mM glutamate puffs. Right- average response before (black) and after (gray) TTX. (**H**) In the ON UBC example, TTX blocked a small tonic inward current but had little effect on the 10 ms 1 mM glutamate puff-evoked current. Right: Average response before (black) and after TTX (gray).

isolate mEPSCs. mEPSCs in UBCs were rare and unusually small and slow (*Figure 5A,B*). Five examples are shown in comparison to a typical granule cell mEPSC (*Figure 5B*). mEPSCs were confirmed to be AMPAR-mediated because they were blocked by AMPAR antagonists GYKI or NBQX (n = 6) (*Figure 5C*). When mEPSCs were present in voltage clamp, their block with GYKI reduced spontaneous EPSPs measured in current clamp (*Figure 5D*). UBC mEPSC amplitudes were 7.20 ± 3.47 pA (mean ± SD, n = 11), less than half the size of mEPSCs recorded from granule cells under the same conditions (18.96 pA ±11.95 pA, n = 8; UBC vs GrC, Mann–Whitney test, p=0.0008, *Figure 5E*). Double exponential functions were fit to the mEPSCs to characterize the decay time of the currents.

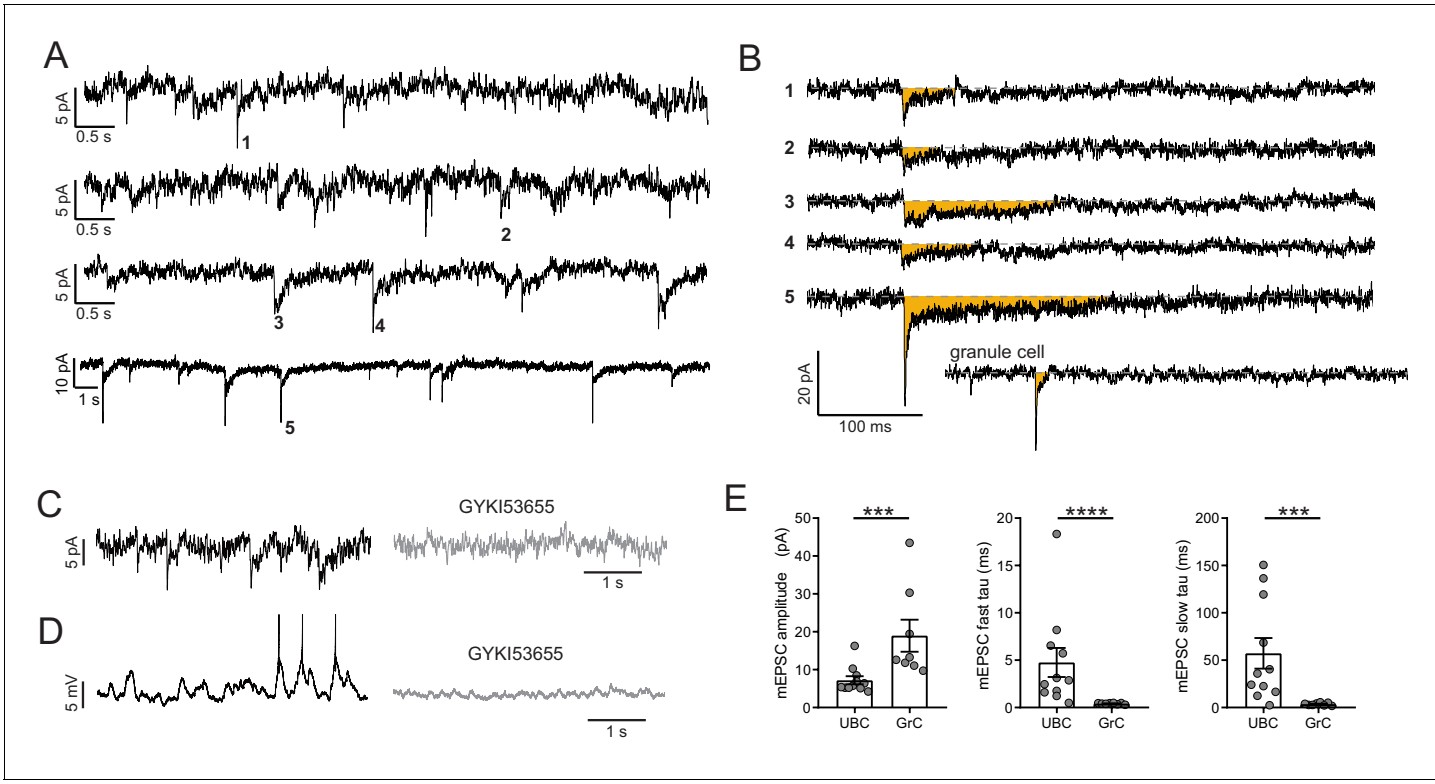

**Figure 5.** mEPSCs suggest vesicular source of ambient glutamate. (**A**) In the presence of TTX, 4 of 15 UBC had mEPSCs that were confidently resolvable above the noise. Each trace is a different cell. Traces are low-pass filtered at 100 Hz, which had a small effect on the amplitude, but was necessary to resolve mEPSCs. (**B**) Example minis indicated by numbers in (**A**). Traces were filtered 2.5 kHz and smoothed (10-point binomial smoothing) to avoid distortion of the peak of the mEPSCs. Note the long duration relative to the granule cell example recorded and filtered in the same way. (**C**) mEPSCs (left) are blocked by GYKI (50 µM) (right). (**D**) The same cell as in (**D**) in current clamp. EPSPs that occasionally reached spike threshold (left) were blocked by GYKI (right). (**E**) mEPSCs from UBCs were smaller in amplitude and slower to decay, quantified by the time constants of double exponential fits. Mann–Whitney tests, n = 11 UBCs, n = 8 GrCs.

Both the fast and slow time constants of the fits to mEPSCs from UBCs were over 10-fold longer than those of GrCs (fast tau, UBC: 4.76 ± 5.11 ms, GrC: 0.41 ± 0.07 ms; slow tau, UBC: 57.23 ± 53.55 ms, GrC: 3.54 ± 1.40 ms. Mann–Whitney test, fast tau, p<0.0001, Mann–Whitney test, slow tau, p=0.0008, *Figure 5E*). In cases where synaptic stimulation was used to identify ON or OFF subtype, only ON UBCs had mEPSCs. The paucity of these events is consistent with the hypothesis that many AMPARs at the ON UBC synapse are displaced from sites of exocytosis, and only sense glutamate from spillover between sites (*Zampini et al., 2016*). Nevertheless, these results confirm that such spontaneous exocytosis occurs.

We tested the hypothesis that vesicular glutamate is the source of glutamate that generates the tonic current by depleting vesicles of glutamate using bafilomycin-A1 (BafA1, 2 µM), which inhibits vacuolar H+ ATPase, preventing loading of synaptic vesicles (*Cavelier and Attwell, 2007*; *Crider et al., 1994*). After slice preparation, the first slice was used for control recordings, while subsequent slices were incubated in 2 µM BafA1 for >1 hr. Because mEPSCs are small in UBCs, the large, frequent mEPSCs of molecular layer stellate cells (*Carter and Regehr, 2002*) were used as positive controls to verify that the BafA1 treatment reduces vesicular glutamate (*Figure 6—figure supplement 1*).

We focused on the tonic inward current generated by AMPAR, as block of desensitization with CTZ would result in a reliable increase in inward tonic current, with the prediction that the enhanced current induced by CTZ would be abolished if vesicular glutamate concentration was depleted. CTZ (100 µM) was applied to controls slices and led to a significant increase in the tonic inward current (−24.3 ± 5.1 pA; p=0.01, n = 12, *Figure 6A,B*), as predicted. Subsequent application of the AMPAR antagonist NBQX (5 µM) blocked the enhanced current. Upon repeating this procedure in ON UBCs in slices incubated in 2 µM BafA1, CTZ still had an effect (−4.16 ± 1.64; p=0.016, n = 13), but nearly sixfold smaller (effect of CTZ in control vs BafA1, p=0.002; n = 25, *Figure 6C*). Thus, the majority of the glutamate driving the tonic current is from synaptic vesicles, likely from the MF terminal.

## Control of synaptic integration by EAATs

UBCs typically respond with an all-or-nothing synaptic response upon electrical stimulation of the white matter of lobe X, despite synchronously activating numerous MF axons, suggesting that a single MF signal is processed by each UBC. What prevents UBCs from sensing nearby MF inputs? In order to address this question, we examined the effect of EAAT blockade on the number of synaptic inputs, as measured by changes in response amplitude following increments in stimulus voltage. A challenge in the experiment is that slow ON or OFF currents are best detected after train stimulation, and at intermediate stimulus voltages, each axon might not be consistently activated during the train. Therefore, single stimuli were delivered after broadening presynaptic action potentials and increasing transmitter release with 1 mM TEA in the bath to block a portion of voltage-gated K-channels (*Ritzau-Jost et al., 2014*). To focus on the response of AMPARs, all other synaptic receptors present in UBCs were blocked and QX314 was added to the pipette solution to prevent postsynaptic spikes.

Of eight UBCs, seven had only a single, sharp increment in EPSC amplitude upon increase in stimulation intensity, suggesting that most ON UBCs only respond to the glutamate released from a single MF (*Figure 7A*). In five of seven UBCs, DL-TBOA revealed a slow spillover EPSC at stimulation intensities below threshold for the fast EPSC that was not apparent before blocking glutamate removal by EAATs (*Figure 7B,C*). After adding DL-TBOA, not only was the response tremendously prolonged, but also the majority of this slow component could be evoked without an accompanying fast EPSC, suggesting that in DL-TBOA, glutamate from a second, more distant MF activates AMPAR on this UBC. Thus, EAATs isolate UBCs from spillover from neighboring MF terminals (*Figure 7D*). By extension, it can be inferred that ambient glutamate is both produced and sensed by a given terminal and is restricted to that synaptic region by EAATs.

## EAAT activity determines the phase delay of the spiking response

The vestibular nerve projecting to lobe X terminates as primary afferent MFs which synapse onto ON UBCs and granule cells (*Balmer and Trussell, 2019*). These MFs fire spontaneously in vivo, and their firing rate is modulated by head motion (*Goldberg and Fernandez, 1971a*; *Goldberg and Fernandez, 1971b*). We developed a stimulation protocol based on *Zampini et al., 2016* to test how

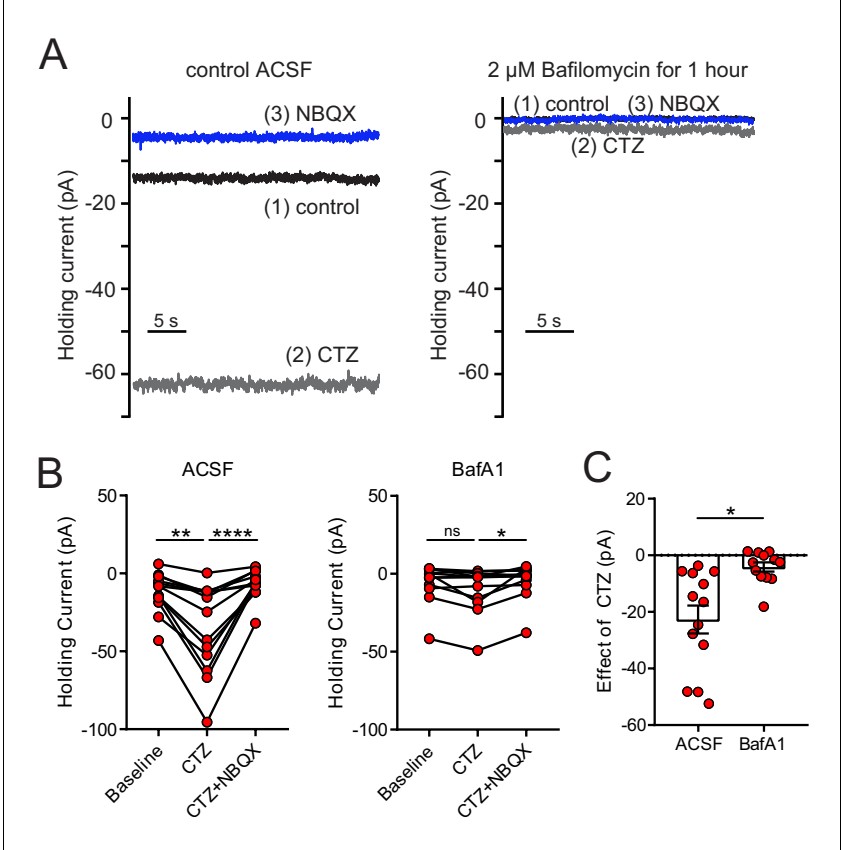

**Figure 6.** Depletion of glutamate from synaptic vesicles confirms vesicular source of ambient glutamate. (**A**) Example recordings of the baseline current of an ON UBC in control (left panel) and of an ON UBC after BafA1 treatment (right), showing the decrease in the current revealed by 100 µM CTZ after treatment with BafA1, with subsequent block by 5 µM NBQX. (**B**) Left: After incubation with ACSF as a control, CTZ increased the inward holding current significantly, which was reversed by NBQX. Friedman test, p<0.0001, n = 12; Dunn's multiple comparisons tests, baseline vs CTZ, p=0.01, CTZ vs CTZ+NBQX, p<0.001. Right- Incubation in BafA1 reduced the effect of CTZ. Friedman test, p=0.0169, n = 12; Dunn's multiple comparisons tests, Baseline vs CTZ, p=0.459; CTZ vs CTZ+NBQX, p=0.0128. (**C**) The effect of CTZ was significantly reduced by incubation in BafA1, unpaired t-test, p=0.002, n = 25.

The online version of this article includes the following figure supplement(s) for figure 6:

**Figure supplement 1.** Control for recordings in the presence of bafilomycin A1.

ON UBCs respond to realistic synaptic input patterns and how glutamate accumulation could affect the resulting spiking responses. After a 10 s, 26 Hz conditioning stimulus, MFs were stimulated at sinusoidally modulated frequencies that approximated the signals they would likely receive during head movement in one direction and then back, with a 1 s cycle (*Figure 8A*). Only cells that responded to every synaptic stimulation without failures were used.

EPSCs in control artificial cerebral spinal fluid (ACSF) (black) showed synaptic depression and a steady-state inward current during the 26 Hz stimulus (*Figure 8A,B*). During frequency-modulated stimulation, EPSCs were apparent during the higher frequency part of the cycle, and these recovered to baseline during the lower frequency part of the cycle (*Figure 8C*). DL-TBOA was applied to increase ambient glutamate by blocking EAATs. After application of DL-TBOA (red), the holding current shifted inward, due to increased tonic AMPAR current (*Figure 8A*). During the 26 Hz stimulus, the steady-state current depressed to a level that was slightly outward relative to pre-stimulus baseline current (*Figure 8B*). During the frequency-modulated stimuli, the current in DL-TBOA was further desensitized, becoming outward during the highest frequencies of the cycle.

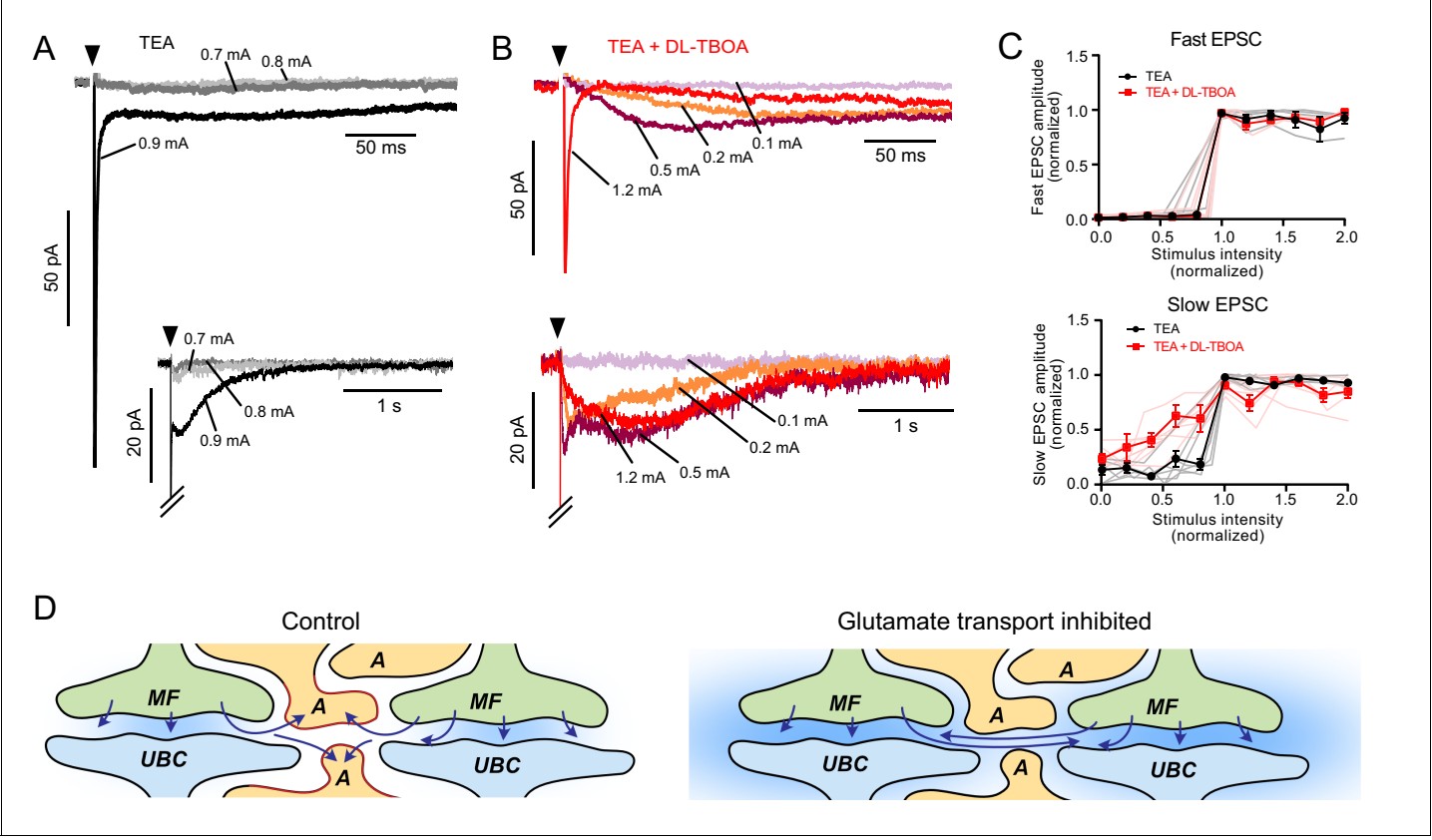

**Figure 7.** EAATs regulate synaptic integration by UBCs. (**A**) Example of AMPAR-mediated EPSC in 1 mM TEA. Top: Single EPSC size (all-or-nothing) at the first stimulation intensity that evoked a current (black). Two stimulation intensities below threshold (light and dark gray) had no EPSC response. Bottom: The same trace with a longer timebase showing the all-or-nothing slow EPSC that occurred at the same stimulation intensity as the fast EPSC. (**B**) Top: Application of DL-TBOA to the same cell as in (**A**) reveals slow EPSCs at stimulation intensities below the level that evoked the fast EPSC (red). Note the reduction in EPSC amplitude due to tonic desensitization. Bottom: In DL-TBOA, the slow EPSCs are much longer than in TEA alone. Without DL-TBOA, only 1 of 11 cells had multiple EPSC levels. In DL-TBOA, five of seven cells had at least two different EPSC sizes, indicating that EAATs prevent AMPARs from detecting multiple mossy fibers during strong synaptic stimulation. (**C**) Top: Fast EPSC amplitudes (relative to the maximal fast EPSC under the same bath conditions) plotted as a function of synaptic stimulation intensity (relative to the lowest intensity that evoked the fast EPSC under the same bath conditions). Both before and after DL-TBOA, the fast EPSC is all-or-nothing. Bottom: Same as above, but showing slow EPSCs. In DL-TBOA, there are multiple amplitude levels. The slow EPSC amplitudes are relative to the maximal slow EPSC in the same bath conditions and the x-axis is relative to the lowest stimulation intensity that evoked the maximal slow EPSC. Error bars mean and SEM. Pink and gray lines are individual experiments. (**D**) Interpretation of these results. In control conditions (left) glutamate is released from the mossy fiber (MF), detected by the receptors on the UBC, then removed from the synaptic cleft by astrocytes (A). When glutamate transporters are inhibited, not only does ambient glutamate build up (blue background), but glutamate released from neighboring synapses is sensed and integrated as slow rising and slow decaying synaptic currents. Thus, glutamate transporters prevent cross-talk between glomeruli and therefore underlie the ability of glomeruli to function as discrete processing units.

In current clamp, DL-TBOA shifted the timing of spiking during the modulated frequency stimulation. In the illustrated example in control ACSF, the cell fired at the beginning of each cycle after the first couple stimuli, slowed during the fastest stimulus rates, and then began firing again as the rate decreased in an offset response (*Figure 8C,D*). In DL-TBOA, the cessation of firing during the frequency increase was nearly complete, picking up markedly when the frequency was lowest (*Figure 8C,D*). To quantify the phase of the offset response, sine wave functions were fit to peristimulus time histograms (PSTHs) generated over cycles of the frequency modulated stimulus (*Figure 8E*). The peak rate of the stimulus was defined as 90°, and the slowest rate was 270°. In control conditions, ON UBCs typically fired both at onset and offset of the stimulus cycle. Because there were two response periods per stimulus cycle, the onset and offset, the sine function fit to the PSTH in control conditions had close to two cycles per stimulus period (1.70 ± 0.29 cycles [mean ± SD]; note the two peaks in the fit to the control PSTH in *Figure 8E*). In DL-TBOA, the sine function fit had

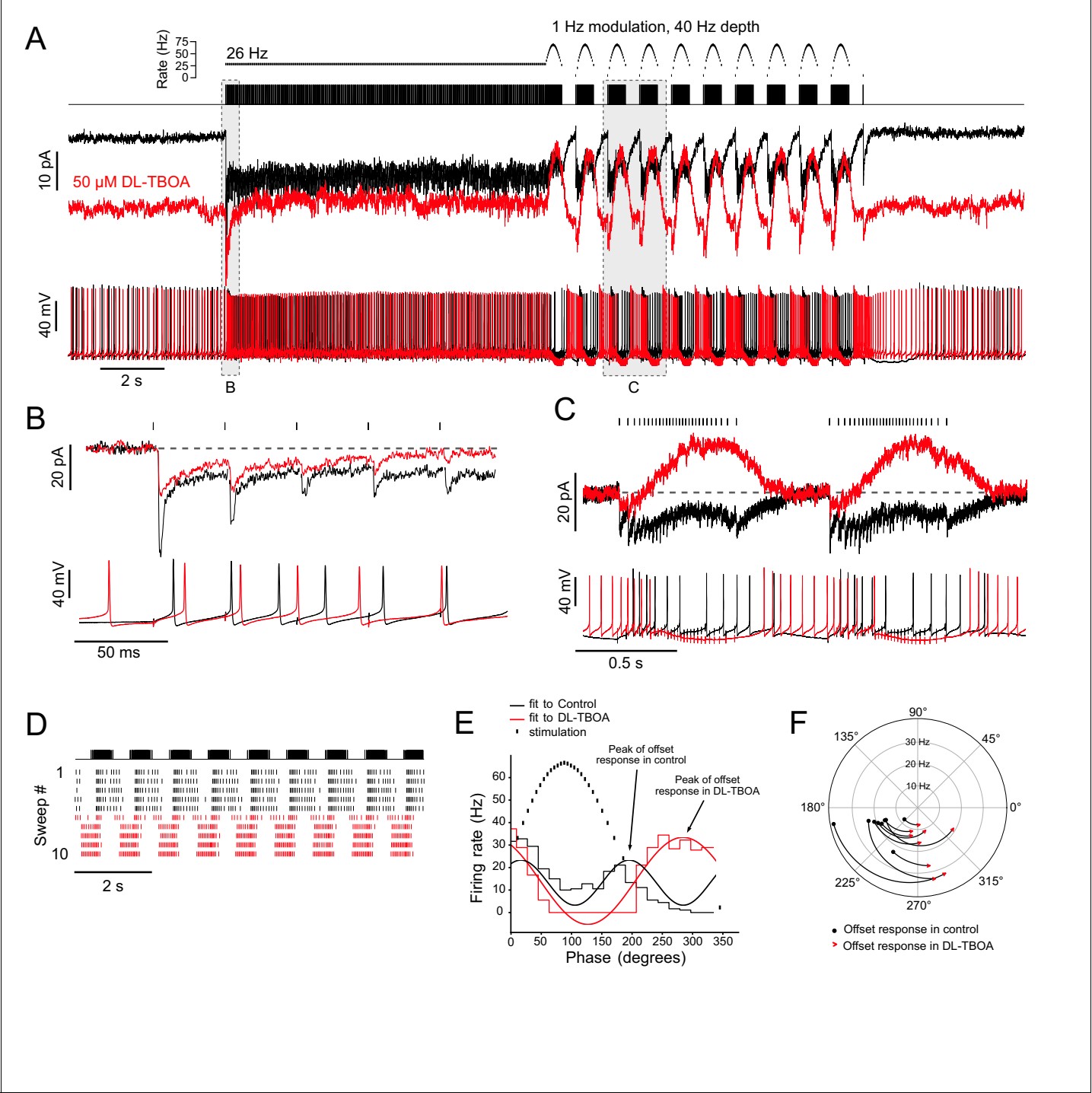

**Figure 8.** Ambient glutamate determines the phase of the spiking response to frequency-modulated synaptic stimulation. (**A**) Example of 1 Hz frequency-modulated synaptic stimulation protocol. Five seconds of no stimulation, followed by 10 s, 26 Hz stimulation, followed by 10 s of modulation with a 40 Hz depth at a modulation rate of 1 Hz. Raster plot shows time of stimuli plotted against instantaneous rate. Below is the stimulus train that was delivered to white matter. Evoked currents are shown for baseline condition (black) and after bath application of DL-TBOA in the same cell (red). DL-TBOA increased inward holding current. Note that during the modulated stimulation the evoked outward current does not go beyond the baseline holding current. Spike trains recorded in current clamp are shown below. Bias current was applied to offset the tonic currents induced by DL-TBOA and to maintain the resting membrane potential at −70 mV. The phase of spiking during the modulated stimulation shifts after block of EAATs. (**B**) Expanded view of the first five pulses of the 26 Hz stimulation. Currents are baseline subtracted. In DL-TBOA, the first EPSC in the train is smaller, indicating tonic desensitization of AMPARs by ambient glutamate. Below, current clamp shows delayed spiking in presence of DL-TBOA. (**C**) Expanded

*Figure 8 continued on next page*

*Figure 8 continued*

view of two cycles of the frequency modulated stimulation. The currents are zeroed right before the increase in stimulation rate. In control, the currents remain net inward during the increased rate of stimulation. In DL-TBOA, desensitization caused a net outward current during the increased stimulation rate. Below, in current clamp, in control conditions, the highest rate of synaptic stimulation caused spiking. In DL-TBOA, the highest rate of stimulation caused hyperpolarization that causes the cell to stop spiking. (**D**) Raster plot showing the application of DL-TBOA starting at sweep 6 (red). Note the shift in timing of the spikes. (**E**) PSTHs during the frequency modulated stimulation were made over several sweeps and cycles. Sine waves were fit to the PSTHs, and their period was used to calculate the phase shift caused by DL-TBOA. In this example the UBC fired during increasing and decreasing stimulus rates. In DL-TBOA, the peak of firing was shifted towards 270° and was continuous from the offset of stimulation until the stimulation rate increases. (**F**) In all cases, DL-TBOA shifted the offset firing response toward the lower stimulation rates, shifting the phase later. The phase of the offset response in control ACSF is represented as a black dot and the phase of the offset response in DL-TBOA of the same ON UBC is represented by the red arrowhead. Firing rate is indicated on the radius. The curves between control and DL-TBOA are arbitrary. These experiments included 5 µM MK-801 in addition to strychnine and SR-95531.

The online version of this article includes the following source code for figure 8:

**Source code 1.** Python script to produce a sinusoidally modulated pattern of TTL pulses that was used to approximate in vivo-like activity patterns.

closer to one cycle per stimulus period (1.10 ± 0.13 cycles), a significant change due to the cell firing during the entire slow stimulation period (paired t-test, p<0.0001, n = 9). Moreover, the peak of the offset response shifted toward the lowest stimulation rate (control: 205.9 ± 15.6° [mean ± SD], DL-TBOA: 282.9 ± 21.5°, paired t-test, p<0.0001, n = 9, *Figure 8F*). This mean 77° shift represents a 214 ms delay. Thus, EAATs controlled the timing of the offset responses of ON UBCs.

## Discussion

### Role of synaptic structure in generating tonic currents and shaping evoked currents

MF terminals form glomerular synapses with dendrites of granule cells, UBCs, and Golgi cells. Glomerular structure and relatively low expression of EAATs allow glutamate spillover in granule cells (*DiGregorio et al., 2002*), slow-rising AMPAR EPSCs (*Nielsen et al., 2004*), and desensitization of AMPARs (*DiGregorio et al., 2007*; *Xu-Friedman and Regehr, 2003*). However, the anatomical and molecular composition of glomeruli differs among target cells, being particularly suited to glutamate accumulation in UBCs. For example, very slow AMPAR EPSCs are a distinctive feature of UBCs as opposed to other MF targets (*Lu et al., 2017*). Accordingly, we observed the tonic glutamate current in UBCs, but not in granule cells. Even mEPSCs were smaller and slower in UBCs vs granule cells. One possibility is that the ultrastructure of synapses formed between MF terminals and granule cells vs UBCs might underlie the observed physiological differences. Electron microscopy reveals very different ultrastructure of these synaptic contacts. The distance between the release sites and the postsynaptic receptors of UBCs may also be greater than granule cells (*Diño et al., 2000a*; *Diño et al., 2000b*; *Rossi et al., 1995*; *Xu-Friedman and Regehr, 2003*). The location of glial cell processes that position EAATs near synapses must almost certainly differ between the cell types, given their different dendritic morphologies. Finally, diffusion barriers specific to the convolutions of the dendritic brush or to extracellular matrix molecules could slow diffusion and increase ambient glutamate.

It is surprising that, despite having enormous postsynaptic densities, UBCs have small-amplitude MF-evoked EPSCs (*Mugnaini et al., 1994*). Our data reveal that one reason for small EPSCs is tonic desensitization caused by ambient glutamate. Relieving desensitization with CTZ increased the fast EPSC amplitude more than twofold. Notably, CTZ also causes an increase in the AMPAR-mediated EPSC of granule cells, without causing a tonic inward current and this difference may be due to the UBC having a dendritic structure that is more restrictive to glutamate diffusion than the granule cell dendrites.

### MFs as the source of ambient glutamate

In the cerebellar granular layer, the only glutamatergic axon terminals are MFs (whether extrinsic or 'intrinsic' MF terminals of the UBCs themselves). The source of ambient glutamate can be most clearly attributed to MF release in experiments where mEPSCs were detected, as they are clearly the result of vesicles being released directly onto postsynaptic AMPA receptors of the recorded

UBC. Exocytic glutamate release from astrocytes remains a possibility, although a controversial one (*Bazargani and Attwell, 2016*). Postsynaptic release of glutamate from the dendrites of UBCs has not been previously described, although the presence synaptic vesicles in UBC dendrites has been reported (*Mugnaini et al., 1994*), and if functional, could provide an additional minor source of glutamate. It should be noted that in our recordings the postsynaptic cell was dialyzed with a glutamate-free solution, which may limit availability of postsynaptic glutamate for vesicles. For these reasons, we favor the simplest interpretation that MFs are the main source of glutamate.

## What concentration of glutamate produces tonic current?

Glutamate receptors vary in their affinities: mGluRs can be activated by relatively low concentrations of glutamate (low micromolar range) (*Conn and Pin, 1997*) and AMPARs, which mediate the majority of phasic excitatory transmission in the CNS, require higher concentrations for maximal activation (~32 µM) (*Lu et al., 2017*; *Raman and Trussell, 1992*). Consistent with their high affinity for glutamate, mGluR2 were tonically activated by ambient glutamate. Indeed, we found that OFF UBCs, whose synaptic responses are dominated by mGluR2 currents, showed tonic inhibition by ambient glutamate.

By contrast, AMPARs desensitize and therefore contribute less to tonic current. ON UBCs had a ~3 pA AMPAR-mediated tonic current, consistent with a ~5 µM steady-state glutamate concentration predicted by our model. Notably, ON UBCs receive primary vestibular afferents that fire constantly at high rates (*Balmer and Trussell, 2019*; *Goldberg and Fernandez, 1971b*). Thus, with a higher level of MF activity, the ambient glutamate concentration may be even higher in vivo.

## Graded transmission

Together with previous studies, our current results suggest that the MF to ON UBC synapse operates quite differently from classical glutamatergic synapses with respect to the buildup and removal of glutamate during synaptic transmission. At conventional glutamatergic synapses, exocytosis raises glutamate levels in the cleft rapidly, and this elevated glutamate is quickly removed by diffusion and uptake or buffering by EAATs (*Clements et al., 1992*; *Diamond and Jahr, 1997*). Thus, the typical postsynaptic cell integrates these extremely brief chemical events to produce stable ongoing responses by generating long-lasting synaptic currents or potentials. As with most synapses, action potentials at presynaptic MF terminals elicit exocytosis and a sharp rise in glutamate at subsynaptic membrane. However, several features of the ON UBC synapse lead to an AMPAR response dictated more by slowly changing levels of transmitter than by postsynaptic integration of brief pulses of glutamate. These features are as follows: (1) persistence of transmitter, (2) receptors not localized immediately beneath the release sites (*Zampini et al., 2016*), (3) cleft structure and EAAT localization favoring accumulation of glutamate and its gradual removal, (4) the entry and recovery of AMPARs from desensitization tracks nonlinearly these gradual changes in transmitter. Thus, postsynaptic currents slowly increase and decay in proportion to changes in the frequency of presynaptic spikes in a nonlinear manner that is governed in part by the activity of EAATs.

## Astrocytes may control the excitability of UBCs by regulating EAATs

Control of ambient glutamate would be expected to profoundly regulate synaptic transmission. One such control point could be EAATs, which are extremely dense on the processes of astrocytes near synapses (*Tzingounis and Wadiche, 2007*). EAAT1–2 contribute to the uptake of glutamate at the UBC synapse, and both are presumed to be expressed by glial cells and not neurons (*Rothstein et al., 1994*). The MF-UBC synapse is exquisitely sensitive to levels of ambient glutamate dictated by EAAT function. By controlling the density or activity of EAATs, astrocytes could regulate both tonic and phasic glutamate currents at this synapse. Astrocytic regulation of EAATs could be a mechanism by which glial cells influence synaptic transmission that does not require release of a transmitter.

Our study also highlighted an additional important role of EAATs besides limiting ambient glutamate and controlling decay time of synaptic responses. In the absence of EAAT activity, spillover between nerve terminals was great enough to produce 'extra' EPSCs in a given neuron. While each UBC had a primary, fast, and slow EPSC from its innervating mossy terminal, blockade of EAATs resulted in additional slow EPSCs likely from transmitter released from terminals on neighboring

UBCs or granule cells (**Figure 7**). These data suggest the possibility that regulation of EAAT expression might alter the effective convergence ratio to each UBC, that is the number of axons producing functional postsynaptic responses for that cell. Such changes could alter multimodal integration of signals in the cerebellum, dependent on the information carried by adjacent nerve terminals.

It should be noted that the data presented here that demonstrate the influence of ambient glutamate on synaptic integration and spiking activity utilized the ability to experimentally increase ambient glutamate. To further explore the role of ambient glutamate at this synapse, technical advances that would allow the experimental reduction of ambient glutamate are warranted, such as overexpression of glutamate transporters or buffers.

### Responses of UBCs to vestibular-like synaptic input

One can predict how UBCs might respond to head movement by examining their firing response to synaptic input that mimics in vivo MF activity. MFs code the velocity of back-and-forth head movements by increasing and decreasing firing rate (**Arenz et al., 2008**; **Goldberg and Fernandez, 1971a**). We find that ON UBCs increase their firing in response to both increases and decreases of the rate of synaptic input and therefore may code for velocity of the head in either direction. In contrast, granule cells receiving the same input would be expected to maintain the velocity signal by increasing and decreasing firing along with the changing rate of the input MF. In addition, the offset response of ON UBCs, which is sensitive to EAAT activity, may delay the parallel fiber vestibular signals in order to coordinate their activity with delayed climbing fiber error signals (**Raymond and Lisberger, 1998**; **Suvrathan et al., 2016**).

## Materials and methods

### Key resources table

| Reagent type (species) or resource | Designation | Source or reference | Identifiers | Additional information |
|---|---|---|---|---|
| Genetic reagent (*M. musculus*) | C57BL/6J | Jackson Laboratory | RRID: IMSR_JAX:000664 | |
| Genetic reagent (*M. musculus*) | B6.TgN(grm2-IL2RA/GFP)1kyo | Dr. Robert Duvoisin (OHSU) PMID:9778244 | RBRC: RBRC01194 | |
| Chemical compound, drug | Alexa Fluor 488 hydrazide sodium salt | ThermoFisher Scientific | Cat # A10436 | |
| Chemical compound, drug | Alexa Fluor 594 hydrazide sodium salt | ThermoFisher Scientific | Cat # A10438 | |
| Chemical compound, drug | GYKI-53655 | Tocris | Cat # 2555 | |
| Chemical compound, drug | JNJ-16259685 | Tocris | Cat # 2333 | |
| Chemical compound, drug | LY-341495 | Tocris | Cat # 1209 | |
| Chemical compound, drug | (+)-MK-801 hydrogen maleate | Sigma | Cat # M107 | |
| Chemical compound, drug | (R)-CPP | Abcam | Cat # ab120159 | |
| Chemical compound, drug | Strychnine hydrochloride | Sigma | Cat # S8753 | |
| Chemical compound, drug | SR-95531 hydrobromide | Tocris | Cat # 1262 | |
| Chemical compound, drug | Tetraethylammonium chloride | Sigma | Cat # 86614 | |
| Chemical compound, drug | NBQX disodium salt | Abcam | Cat # ab120046 | |
| Chemical compound, drug | Cyclothiazide | Tocris | Cat # 0713 | |
| Chemical compound, drug | Bafilomycin A1 | InvivoGen | Cat # tlrl-baf1 | |
| Chemical compound, drug | DL-TBOA | Tocris | Cat # 1223 | |
| Chemical compound, drug | TFB-TBOA | Tocris | Cat # 2532 | |
| Chemical compound, drug | Tetrodotoxin citrate | Tocris | Cat # 1069 | |
| Software, algorithm | pClamp 10 | Molecular Devices | RRID:SCR_011323 | |
| Software, algorithm | Igor Pro 8 | WaveMetrics | RRID:SCR_000325 | |

*Continued on next page*

*Continued*

| Reagent type (species) or resource | Designation | Source or reference | Identifiers | Additional information |
|---|---|---|---|---|
| Software, algorithm | Prism 8 | GraphPad | RRID:SCR_002798 | |
| Software, algorithm | Excel | Microsoft | RRID:SCR_016137 | |
| Software, algorithm | Axograph X | Axograph | RRID:SCR_014284 | |
| Software, algorithm | NEURON | PMID:9248061 | RRID:SCR_005393 | |
| Software, algorithm | Affinity Designer | Serif | RRID:SCR_016952 | |
| Software, algorithm | Python | python.org | RRID:SCR_008394 | |
| Software, algorithm | NumPy | numpy.org | RRID:SCR_008633 | |
| Software, algorithm | MatPlotLib | matplotlib.org | RRID:SCR_008624 | |

## Animals

Animals used in this study were primarily C57BL/6J-TgN(grm2-IL2RA/GFP)1kyo line. In this mouse line, GFP is tagged to human interleukin-2 receptor α subunit with expression driven by an mGluR2 promoter (*Watanabe et al., 1998*). Both ON and OFF UBCs express mGluR2, thus both subsets are labeled in this line (*Nunzi et al., 2002*), with GFP targeted to the plasma membrane. They were bred in a colony maintained in the animal facility managed by the Department of Comparative Medicine, and all procedures were approved by the Oregon Health and Science University's Institutional Animal Care and Use Committee. All experiments were performed in brain sections from males and females, postnatal days 21–35 (P21–P35). Synapse formation between the presynaptic MF terminal and the postsynaptic dendritic brush structure of the UBCs is mature in animals older than P21 (*Hámori and Somogyi, 1983*; *Morin et al., 2001*). Transgenic mice were phenotyped by light at P0–P3.

## Brain slice preparation

Animals were anesthetized with isoflurane, decapitated, and the cerebellum was dissected from the skull under ice-cold high-sucrose ACSF solution containing the following (in mM): 87 NaCl, 75 sucrose, 25 $NaHCO_3$, 25 glucose, 2.5 KCl, 1.25 $NaH_2PO_4$, 0.4 Na-ascorbate, 2 Na-pyruvate, 0.5 $CaCl_2$, 7 $MgCl_2$, and 5 μm R-CPP, bubbled with 5% $CO_2$/95% $O_2$ (*Bischofberger et al., 2006*). Parasagittal cerebellum sections 300 μm thick were cut with a vibratome (VT1200S, Leica or 7000smz-2, Campden Instruments) in ice-cold high-sucrose ACSF. Slices recovered at 35°C for 30–40 min, in ACSF containing the following (in mM): 130 NaCl, 2.1 KCl, 1.2 $KH_2PO_4$, 3 Na-HEPES, 10 glucose, 20 $NaHCO_3$, 0.4 Na-ascorbate, 2 Na-pyruvate, 2 $CaCl_2$, 1 $MgSO_4$, bubbled with 5% $CO_2$/95% $O_2$ (300–305 mOsm). In the experiments that attempted to approximate in vivo-like stimulation, 1.2 mM Ca was used. In some experiments, 5 μm R-CPP or MK-801 was included in high-sucrose ACSF. Slices were kept at room temperature (~23°C) until recording. Recordings were performed from cerebellar lobe X, within 6 hr of preparation.

## Electrophysiological recordings

During recordings, slices were perfused with recording ACSF using a peristaltic pump (Ismatec) at 2–3 ml/min and maintained at ~34°C with an inline heater (SH-27B, Warner Instruments). All experiments were done in the presence of 5 μM SR-95531 and 0.5 μM strychnine. The recording set up was composed of a Zeiss Axioskop 2 FS Plus microscope with Dodt gradient contrast optics (*Dodt et al., 2002*), with 10× and 60× (water immersion) Olympus objectives. UBCs were initially identified by soma size or GFP fluorescence, but intracellular recording solution always contained 5 μM Alexa Fluor 594 hydrazide sodium salt (Life Technologies). Thus, in whole-cell mode, UBCs were easily identifiable by morphology after dye dialysis. Patch electrodes were pulled with borosilicate glass capillaries (OD 1.2 mm and ID 0.68 mm, World Precision Instruments), with an upright puller (PC10, Narishige).

For whole-cell recordings, intracellular recording solution contained (mM): 113 K-gluconate, 9 HEPES, 4.5 $MgCl_2$, 0.1 EGTA, 14 Tris–phosphocreatine, 4 $Na_2$-ATP, 0.3 Tris–GTP (adjusted to 290 mOsm with sucrose), pH 7.2–7.25. All recordings were corrected for a −10 mV junction potential. For data acquisition, we used a Multiclamp 700B amplifier and pClamp nine software (Molecular

Devices). Signals were sampled at 20–50 kHz using a Digidata (1440A, Molecular Devices) analog–digital converter board. Current signals in voltage clamp were acquired with 5–10× gain and low-pass filtered at 10 kHz, with further filtering applied offline. Stimulation artifacts have been removed for clarity. Patch pipette tip resistance was 5–8 MΩ; series resistance was compensated with correction 20–40% and prediction 50–70%, bandwidth 2 kHz. Membrane potential was held constant at −70 mV in voltage-clamp experiments.

Electrical stimulation was performed using a concentric bipolar electrode (CBBPC75, FHC) placed in the white matter of the sagittal cerebellar slice. Some fraction of synaptic responses may be due to the activation of MF terminals of UBCs' axons, in addition to the extrinsic MFs that make up the majority of the white matter. Stimuli were evoked using a stimulus isolation unit (Iso Flex, A.M.P.I.) delivering 100–250 µs duration pulses of 0–90 V. Frequency-modulated stimulation protocols were generated using custom Python scripts (*Figure 8—source code 1*) approximating those used by *Zampini et al., 2016*. The stimulation train consisted of a 10 s 26 Hz train, followed by a 10 s train with a modulation frequency of 0.3, 1, or 3 Hz and a modulation depth of 6, 40, or 120, respectively. The stimulus was calculated by:

$$(sin\{2\pi * (carrier * t)\} + modulator) * (d/f)$$

where *modulator* = $sin(2\pi ft) * d$, carrier = 26 Hz, *f* is modulation frequency, *d* is modulation depth, and *t* is time. As the purpose of these experiments was to determine the effect of ambient glutamate on synaptic currents and resulting spiking patterns, bias current was injected to maintain the resting membrane potential around −70 mV throughout the experiment.

Puff application of agonists and antagonists was delivered through a Picospritzer II (Parker Instrumentation), at 5–10 psi, with borosilicate glass capillaries. Glutamate applications were at 1 mM and 7–10 ms in duration and were always used prior to recordings for subtype identification. Puff application of control solution without drugs ruled out puff artifacts. The puff pipette was kept >20 µm away from the dendritic brush and soma to avoid mechanical disturbance of the cell.

## AMPA receptor kinetic model

The AMPAR kinetic model was constructed by fitting the reaction rates of a 13-state scheme to AMPAR currents from whole-cell recordings of dissociated UBCs (*Lu et al., 2017*) using Axograph X software (RRID:SCR_014284). Parameters (*Figure 1—figure supplement 2—source data 1*) were adjusted to account for the steady-state dose–response curve, rate of decay of the fast initial EPSC in a train, the rate of decline of the amplitude of fast EPSCs during the train, the relative amplitude of the peak fast EPSC vs the peak slow current after the train, and the rise and decay time of the slow current. The key feature yielding a non-monotonic dose–response relationship was the differences in the rates governing entry and exit of D1-2 vs D3-4. Alterations in rates of opening and closing of the different open states could also generate a fall-off in current at high agonist concentrations, but also distorted fast deactivation to an unacceptable degree.

The AMPAR kinetic model was implemented using NMDL in NEURON (*Balmer and Trussell, 2021*; *Carnevale and Hines, 2006*; *Hines and Carnevale, 2000*) (RRID:SCR_005393). AMPAR conductance (*g*) was calculated from the occupancy of four open states (*Figure 1—figure supplement 2—source data 1*). AMPAR currents were calculated at a holding potential (*v*) of −70 mV and a reversal potential (*Erev*) of 0 mV to match experimental conditions, using the following equation:

$$AMPA\ current = g * (v - Erev)$$

In order to use the model that was derived from recordings made at room temperature to approximate synaptic currents recorded in slices at near physiological temperatures, the rates were adjusted. We chose to simply apply a Q10 (temperature coefficient) of 2.4 for all rates in the kinetic model for the following reasons: Multiplication by a single factor that equally affected all transition rates between states of an AMPAR model was sufficient to fit temperature-dependent changes of AMPAR currents at the Calyx of Held (*Postlethwaite et al., 2007*). Association and dissociation rate constants are temperature independent for acetylcholine receptors, which are similar in structure to AMPARs (*Gupta and Auerbach, 2011*). By increasing both the binding and unbinding constants by the same value, the association and dissociation rate constants remain unchanged, but the

transitions between states are sped. This increased the rise and decay rates of fast synaptic currents and had no effect on the dose–response curves.

## Glutamate transient prediction

Glutamate time courses were generated and used to stimulate the model synapse. The resulting currents were fit to experimentally isolate AMPA currents from whole-cell recordings of UBCs in brain slices. This approach predicts the glutamate concentration profiles that may have occurred at the AMPARs at the UBC synapse. Glutamate transients were generated using a 3D diffusion equation:

$$[glu] = \frac{M}{8\alpha\left\{\pi\left(D/\lambda^2\right)t\right\}^{3/2}}\exp\left\{\frac{-r^2}{4\left(D/\lambda^2\right)t}\right\} + ambient$$

where glutamate concentration [glu] evolves over time, t. Moles of glutamate released, M, and 3D distance between release site and receptors, r, and the level of ambient glutamate (ambient) were varied across fitting trials. The other variables remained constant: volume fraction ($\alpha$ = 0.21) (**Nicholson and Phillips, 1981**), diffusion coefficient (D = 0.33 cm$^2$/s) (**Nielsen et al., 2004**), and tortuosity ($\lambda$ = 1.55) (**Barbour and Häusser, 1997**; **Nicholson and Phillips, 1981**). The diffusion profile generated by this equation has a slowly decaying tail that is characteristic of a hyperbolic curve, which in this application approximates the slow exit of glutamate out of the synaptic cleft. Although the equation produced synaptic current models that fit the experimental data well, it remains a rough approximation due to assumptions of a single point of release and single point of detection (see **Balmer and Trussell, 2021** for model code). Glutamate time courses from individual release events were added linearly to generate trains. No presynaptic depression was assumed. The other parameter of the model that was varied to optimize the fit was the maximal conductance of the AMPARs, which approximates the number of receptors present. The fitting procedure returned an average value for r of 1.23 ± 0.36 µm, which includes receptors located both near sites of exocytosis and also far (and therefore only activated by pooling of transmitter). This result is consistent with the proposal of **Zampini et al., 2016** that many receptors at this synapse are activated by glutamate diffusing some distance from sites of release.

The leak current (baseline) of the experimentally recorded trace was also adjusted to correct for leak channels not implemented in the model that may have been active in the cell, as well as other typical sources of leak that accompany whole-cell recordings. Experimental traces were averaged across 5–10 trials, low-pass filtered (1 kHz), and the stimulation artifacts were blanked. Simulated glutamate transients were optimized by least-square fitting the simulated synaptic current to experimentally isolated AMPAR currents using NEURON's Praxis package (RRID:SCR_005393).

## Acknowledgements

This study was funded by National Institutes of Health (NIH) Grants NS028901, R35NS116798, and DC004450 (PI: LOT); F31 DC012454 and NL Tartar Trust Fellowship (PI: CBM); F32 DC014878, K99 DC016905, and Hearing Health Foundation Emerging Research Grant (PI: TSB); and P30 DC005983 (PI: Gillespie, Peter). We would like to thank Dr. Craig Jahr and members of the Trussell Lab for helpful discussions.

## Additional information

### Funding

| Funder | Grant reference number | Author |
| --- | --- | --- |
| National Institute of Neurological Disorders and Stroke | NS028901 | Laurence O Trussell |
| National Institute of Neurological Disorders and Stroke | NS116798 | Laurence O Trussell |
| National Institute on Deafness and Other Communication Disorders | DC004450 | Laurence O Trussell |

| National Institute on Deafness and Other Communication Disorders | DC016905 | Timothy S Balmer |
|---|---|---|
| National Institute on Deafness and Other Communication Disorders | DC014878 | Timothy S Balmer |
| National Institute on Deafness and Other Communication Disorders | DC012454 | Carolina Borges-Merjane |

The funders had no role in study design, data collection and interpretation, or the decision to submit the work for publication.

## Author contributions

Timothy S Balmer, Conceptualization, Data curation, Software, Formal analysis, Funding acquisition, Validation, Investigation, Visualization, Writing - original draft, Writing - review and editing; Carolina Borges-Merjane, Conceptualization, Formal analysis, Investigation, Methodology, Writing - review and editing; Laurence O Trussell, Conceptualization, Resources, Supervision, Funding acquisition, Writing - original draft, Project administration, Writing - review and editing

## Author ORCIDs

Timothy S Balmer ![ORCID] https://orcid.org/0000-0002-8864-5465
Carolina Borges-Merjane ![ORCID] https://orcid.org/0000-0003-0005-401X
Laurence O Trussell ![ORCID] https://orcid.org/0000-0003-1171-2356

## Ethics

Animal experimentation: All experiments were performed under the approval of the institutional animal care and use committee (IACUC) of Oregon Health and Science University, assurance #A3304-01.

## Decision letter and Author response

Decision letter https://doi.org/10.7554/eLife.63819.sa1
Author response https://doi.org/10.7554/eLife.63819.sa2

# Additional files

## Supplementary files

• Transparent reporting form

## Data availability

All data generated or analysed during this study are included in the manuscript and supporting files.

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
