## [Decision Letter]

**Acceptance summary:**

This paper provides convincing evidence that a physiologically significant level of ambient glutamate is present at a synapse in the cerebellum, where it controls spontaneous firing and regulates phasic postsynaptic currents. The ambient glutamate is the result of spontaneous vesicle fusion and not action potential-evoked release. These synapses are virtually always exposed to a significant level of glutamate unlike classical fast synapses, which has a powerful impact on neuronal firing patterns. This regulatory mechanism endows these synapses with unique functional properties.

**Decision letter after peer review:**

Thank you for submitting your article "Incomplete removal of extracellular glutamate controls synaptic transmission and integration at a cerebellar synapse" for consideration by *eLife*. Your article has been reviewed by three peer reviewers, one of whom is a member of our Board of Reviewing Editors, and the evaluation has been overseen by John Huguenard as the Senior Editor. The following individual involved in review of your submission has agreed to reveal their identity: Alain Marty (Reviewer #2).

The reviewers have discussed the reviews with one another and the Reviewing Editor has drafted this decision to help you prepare a revised submission.

In the current study Balmer et al. showed that ambient glutamate in the synaptic cleft tonically activates ionotropic and metabotropic receptors in UBC cells in the cerebellum. This is quite different from the typical glutamatergic synapse where synaptic responses rapidly rise and fall. Ambient glutamate responses described in the current manuscript exist in the absence of action potentials in the presynaptic neurons and result from spontaneous vesicular glutamate release. The level of ambient glutamate, which is set by glutamate transporters in the synaptic cleft, endows the neurons with a complex signalling mechanism via the regulation of the length and polarity of postsynaptic responses. The observed change in postsynaptic currents is much slower than changes in the frequency of presynaptic spikes. The authors illustrate the functional complexity of this synaptic reposes by showing that ambient glutamate can shift the phase of responsiveness of UBC cells to periodic stimulation.

As you will see in the detailed reviews below, all three reviewers were unanimous to praise the rigorous experimental design and the high quality of the experimental data. The authors' conclusions are solidly built on their findings and fully supported by the data presented. While there is general enthusiasm for this study, the modelling data presented on Figure 2. was identified as the weakest point of the manuscript. Reviewers raised several critical points related to model. They also find that this is not a critical element of the study and suggest that the authors either move this figure to a supplementary figure or remove it completely.

Reviewer #1:

In this manuscript Balmer et al. investigates the effect of extracellularly accumulating glutamate at MF-UBC synapses in the cerebellum. The authors found that glutamate accumulating in the synaptic cleft influenced the spontaneous firing rate of OFF cells and desensitized AMPA receptors in ON cells. They identified the source of ambient glutamate as spontaneous, AP-independent release from MF terminals. mGluR2-medated tonic current was present in OFF cells, and tonic desensitization of AMPA currents was observed in ON cells.

The study investigates differences in ON and OFF cells and identifies dominant glutamate mediated tonic currents in both groups suggesting the presence of ambient glutamate at these synapses. The authors also show that these tonic currents could alter firing properties and hence could significantly impact downstream network function.

The study offers novel insights into the presence of tonic currents in UBCs mediated by ambient glutamate. This is an important component of the functional landscape shaping firing patterns in this network. The study is carefully planned and executed.

The focus of the study is the presence and role of tonic currents that are mediated by ambient glutamate that is the result of AP-independent release. There is an inherent logical difficulty here since the presence of these tonic currents have been convincingly demonstrated, but the “ideal” experiments investigating the role of these currents (i.e. isolating and selectively removing tonic currents) in synaptic integration and spiking is technically very difficult. The authors use the TBOA experiments in their functional experiments, but these experiments will only allow them to draw conclusions about the role of EAATs in signal integration without having direct insights into the role of ambient glutamate in the cleft. This disconnect is expected, but the manuscript could be improved to make this issue clearer to the reader.

The study is a logical continuation of the authors previous papers (Balmer et al., 2017 and Borges-Merjane and Trussel, 2015). This means that there is some unavoidable overlap between the experiments even if the focus is quite distinct here. The Discussion could benefit from a more in-depth investigation of how the current and previous findings combine at the functional level. For example, it would be interesting to see a detailed discussion on the relationship between the CTZ experiments presented here on Figure 4 and uncaging experiments on Figure 3 of Balmer et al.

The authors suggest that the most likely source of glutamate driving the tonic currents is from MF terminals. The authors should elaborate on the level of certainty they have in this and outline the alternatives.

UBCs terminate not only on granule cells but also on other UBCs, potentially forming a chain of interconnected UBCs (Dorp and De Zeeux, 2015). Could the synaptic inputs investigated by the authors be disynaptic?

Figure 3A shows an extreme example, the average tonic AMPA current is ~3pA and this small current is explained by AMPA-R desensitization. One of the authors' point in the Discussion is about how desensitization leads to a smaller tonic current. May be a more typical example in Figure 3A would better illustrate the main findings.

Figure 8 is focused on ON cells. In these cells tonic currents seems to be rather small due to AMPA-R desensitization, while the mGluR2-medated tonic outward current in OFF cells is larger. What happens in OFF cells?

Figure 8 The firing of the neuron is a result of interactions between synaptic and intrinsic conductance. Blockade of EAATs with TBOA (and therefore increase in ambient glutamate) affects both a tonic current and the evoked synaptic current. Can the authors tease apart these two effects? In presence of TBOA, the baseline firing of the neuron in absence of synaptic stimulation is increased due to augmented inward tonic current. First, it would be important to know how the baseline membrane potential was adjusted (or were recordings performed at resting Vm?). Second, how much did the baseline firing change after TBOA application, due to the increased tonic inward current? Third, and most importantly, the authors should test whether artificially increasing the baseline firing of the neuron through somatic current injection (and recapitulating the effects of TBOA on basal firing rates) can mimic the effect of TBOA on the firing rate during the synaptic stimulation protocol. Overall, this would provide reassurance that all effects can be attributed to EAATs blockade as the authors suggest.

Figure 6F, right panel: the NBQX trace is almost completely hides the control trace. Perhaps use the same indicators [(1) control, (2) CTZ, (3) NBQX] as on the left panel.

Reviewer #2:

In recent years the Trussell group has produced several interesting papers examining the mechanisms of synaptic transmission in MF-UBC synapses. In the present work, it is shown that under basal conditions, there is enough ambiant glutamate to significantly activate ionotropic and metabotropic receptors at these synapses. Furthermore, the authors convincingly show that the basal glutamate is provided by the continuous supply linked to spontaneous vesicular release from resting MF terminals. Finally, in a spectacular final figure, the authors present intriguing data suggesting that ambient glutamate may shift the phase of responsiveness of the UBC cells to periodic stimulation. This leads to the very interesting suggestion that the UBC to GC relay may shift the phase of the incoming cerebellar sensory input in a flexible manner.

The design and execution of the experiments are excellent, and the conclusions of the paper are both novel and justified.

My only substantial criticism concerns the treatment of glutamate diffusion in the glomerulus. In both Introduction and Discussion, the authors say that glutamate escape out of the glomerulus may occur slowly, on a time scale of 100s of ms. But their actual diffusion model, that is used in the Results section, does not incorporate a separate, slow phase representing glutamate exit out of the glomerulus. Rather, it treats glutamate diffusion as a single process that is homogeneous in space, with global slowing factors reflecting volume fraction and tortuosity. This results in an apparent contradiction that should somehow be addressed, at least with a caveat sentence when presenting the diffusion model, saying that this model is only a rough approximation.

I have listed below specific suggestions that should be taken into consideration when revising the manuscript.

1) Glutamate diffusion model:

Whereas the AMPAR activation is modelled in detail (Figure 2A), the diffusion of glutamate is described by a highly simplified model (diffusion equation). This model does not attempt to incorporate the complicated MF terminal geometry, which is understandable. However, it may be oversimplified, as all glutamate sources are considered equivalent, and the diffusion process is homogeneous in space.

Incorporating crevices in the model, as was done by Nielsen et al. (2004), results in the prediction of two phases of glutamate concentration decay, with a slow phase reflecting glutamate escape once the glutamate concentration has equilibrated within the glomerulus. A biphasic glutamate decay model was likewise developed by Barbour et al. (1994) at PF-PC synapses. Such a biphasic decay could help interpreting the time course of mEPSCs. Specifically, it is possible that the slow time constant of mEPSC decay could reflect a slow component of glutamate concentration decay (escape out of the glomerulus), rather than a low value for channel resensitization rates, as it presently appears from Supplemental Table 1.

The possibility of a slow glutamate escape would give a likely mechanistic interpretation for the finding of a significant basal glutamate concentration inside the synaptic cleft. By contrast, this conclusion of the paper is presumably difficult to reconcile with the monotonic glutamate decay assumed, given the scarcity of recorded mEPSCs.

Again, I do not suggest developing a realistic diffusion model based on glomerulus morphology: this would be a huge task that would require the introduction of many geometric parameters that are presently unknown. But the authors should acknowledge in their modelling section the possibility that glutamate concentration decay could be biphasic, and that this could be an explanation for the slow phase of mEPSC decay.

2) mGluR1 activation by basal glutamate?

When reading the section on effects of basal glutamate in ON UBCs, the question arises as to whether mGluR1s would be activated in addition to AMPARs. Was this examined? Even if the results were negative, it would be good to mention them for completeness.

3) Figure 6:

This figure has two parts: panels A-E describe mEPSCs in UBCs, and F-H describe combined bafilomycin/CTZ suggesting that the source of ambient glutamate is vesicular. It is the second part which is most directly relevant to the paper. While describing mEPSCs is valuable -and may yield important clues on glutamate diffusion, see above-, the fact that mEPSCs are present in these cells does not represent a surprise. In view of the above the authors should consider keeping only parts F-H as main text figure, and downgrading parts A-E as an additional supplementary figure.

4) Model of Figure 8:

In the analysis of Figure 8E-F, the control response is modelled as a second harmonic of the MF spike modulation, while the response in TBOA is modelled as a first harmonic. But looking at the data in D-E, it appears that modelling the control response as a second harmonic is rather arbitrary. The amplitude of the second wave is much smaller than that of the first one, indicating that the true period remains the driving period. Looking at the raw data in D, it appears that the black spikes are roughly synchronized with the driving signal, except for a small delay, whereas the red spikes are almost opposite in phase with respect to the driving signal. This interpretation could be conveyed by performing a first harmonic fit to the black data as well as to the red data in E. Fitting the black histogram in E with a first harmonic would considerably simplify the presentation of the next panel (F). Presently panel F is almost impossible to understand. It is unclear what the radius in F represents (probably the firing rate for the second harmonic was doubled; but this is hard to justify). It is also unclear how the arc circles joining black and red dots were obtained. And it is unclear why the phase chosen for the black dots corresponds to the second peak in E rather than to the first one.

Reviewer #3:

This study shows how glutamate transporters limit activation of glutamate receptors at ON and OFF UBCs. The experiments show quite convincingly that glutamate transporters regulate ambient glutamate levels and synaptic glutamate diffusion dynamics. ON and OFF UBCs serve as controls for each other in various experiments, enabling the authors to distinguish effects on ambient and evoked transmitter levels. Further experiments show that ambient glutamate is due to spontaneous vesicle release and that transporters influence physiologically relevant UBC signaling. The experiments appear rigorously performed and are clearly presented. My primary concern is with the kinetic model used to estimate the glutamate concentration.

The authors estimate the glutamate concentration time course using a kinetic model of AMPA receptors. The experimental bases for model parameters are not described thoroughly and raise many questions. The authors note that parameters were tuned to match whole-cell responses recorded from dissociated UBCs (Lu, et al., 2014), experiments in which glutamate was applied to the whole cell with relatively slow exchange times (estimated by the authors to be 50 ms). These methods are likely inadequate to constrain many of the parameters in the model. As just one example, the authors determine that the glutamate binding rate at room temperature to a single binding site is 2×108 s^-1^M^-1^s^-1^, which is certainly within theoretical diffusion limits but, as far as I know, is 10 times faster than any previous estimates and cannot be derived reliably from the cited whole-cell data. This is likely to be an important issue, because a higher binding rate would confer higher sensitivity in non-equilibrium conditions. I do not suggest that the authors undertake an entire kinetic analysis of UBC AMPA receptors, but I would ask that they add more detail supporting the rate constants and how much the model's behavior (beyond the non-monotonic dose-response) relies on specific values. I would also suggest that the model constitutes the weakest part of the paper and distracts somewhat from an otherwise pleasing flow of nice physiological experiments. I would suggest that the model is not necessary and, if retained, might function better as a discussion point at the end.

---

## [Author Response]

As you will see in the detailed reviews below, all three reviewers were unanimous to praise the rigorous experimental design and the high quality of the experimental data. The authors' conclusions are solidly built on their findings and fully supported by the data presented. While there is general enthusiasm for this study, the modelling data presented on Figure 2. was identified as the weakest point of the manuscript. Reviewers raised several critical points related to model. They also find that this is not a critical element of the study and suggest that the authors either move this figure to a supplementary figure or remove it completely.

We have opted to move Figure 2 to a supplemental figure. While the points the reviewers made are well taken, we feel the modeling approach was key. Notably, we are not using the model as a confirmation of experiments, but rather to generate experimental testable hypotheses. The model suggested the presence of ambient glutamate and the paper then goes on to confirm the model in that regard. We hope that by positioning it as an early supplement, we honor both the reviewers concerns and our own intentions.

Reviewer #1:[…] The authors use the TBOA experiments in their functional experiments, but these experiments will only allow them to draw conclusions about the role of EAATs in signal integration without having direct insights into the role of ambient glutamate in the cleft. This disconnect is expected, but the manuscript could be improved to make this issue clearer to the reader.

This is a good point. We have added a caveat in the text.

The study is a logical continuation of the authors previous papers (Balmer et al., 2017 and Borges-Merjane and Trussel, 2015). This means that there is some unavoidable overlap between the experiments even if the focus is quite distinct here. The Discussion could benefit from a more in-depth investigation of how the current and previous findings combine at the functional level. For example, it would be interesting to see a detailed discussion on the relationship between the CTZ experiments presented here on Figure 4 and uncaging experiments on Figure 3 of Balmer et al.

We appreciate that the reviewer has recognized the continuity of these works and have added text, to succinctly connect the present work to the conclusions of the uncaging study.

The authors suggest that the most likely source of glutamate driving the tonic currents is from MF terminals. The authors should elaborate on the level of certainty they have in this and outline the alternatives.

We have added an exploration of the alternatives to the Discussion section.

UBCs terminate not only on granule cells but also on other UBCs, potentially forming a chain of interconnected UBCs (Dorp and De Zeeux, 2015). Could the synaptic inputs investigated by the authors be disynaptic?

The reviewer makes an interesting point. We doubt the dataset we show here includes such cases. Where disynaptic responses were apparent, they were characterized by longer delays and jitter in the onset of the fast AMPA EPSC. If the stimulus electrode in the white matter was activating UBC soma, the trains of responses would almost certainly trigger spike bursts, and therefore bursts of EPSCs. One possibility is that the electrode stimulated UBC axons that had been severed from their soma, but these are not so likely to populate the white matter containing cerebellar input fibers. We make a note about this now in the Materials and methods.

Figure 3A shows an extreme example, the average tonic AMPA current is ~3pA and this small current is explained by AMPA-R desensitization. One of the authors' point in the Discussion is about how desensitization leads to a smaller tonic current. May be a more typical example in Figure 3A would better illustrate the main findings.

Since a more typical response is very small and noisy, we chose to show an outlier and label it as such.

Figure 8 is focused on ON cells. In these cells tonic currents seems to be rather small due to AMPA-R desensitization, while the mGluR2-medated tonic outward current in OFF cells is larger. What happens in OFF cells?

We chose to focus on ON cells because in the rare cases where we had an OFF cell for this kind of experiment, the results were frankly not very interesting: the cell remained off for most of the during of stimulation, particularly in TBOA.

Figure 8 The firing of the neuron is a result of interactions between synaptic and intrinsic conductance. Blockade of EAATs with TBOA (and therefore increase in ambient glutamate) affects both a tonic current and the evoked synaptic current. Can the authors tease apart these two effects? In presence of TBOA, the baseline firing of the neuron in absence of synaptic stimulation is increased due to augmented inward tonic current. First, it would be important to know how the baseline membrane potential was adjusted (or were recordings performed at resting Vm?). Second, how much did the baseline firing change after TBOA application, due to the increased tonic inward current? Third, and most importantly, the authors should test whether artificially increasing the baseline firing of the neuron through somatic current injection (and recapitulating the effects of TBOA on basal firing rates) can mimic the effect of TBOA on the firing rate during the synaptic stimulation protocol. Overall, this would provide reassurance that all effects can be attributed to EAATs blockade as the authors suggest.

“Can the authors tease apart these two effects?”. These two are manifestations of a single effect, which is activation of the receptors by accumulated glutamate. Thus, they are not separable. For mGluR2, the receptors are tonically activated thus generating steady current and in parallel reducing the fraction available for activation by the evoked response. For AMPAR, activation and desensitization.

“…know how the baseline membrane potential was adjusted”. We added a sentence to Materials and methods to indicate that bias current was applied to keep the resting membrane potential constant. Small currents, on the order of 20 pA would be sufficient to lead to depolarization block. We did not do the experiment suggested, to inject current to mimic TBOA, but it should be clear that since bias current was injected, the effects we described must be related to the nonlinear responses of the AMPAR to ambient glutamate and not merely a steady current buildup.

Figure 6F, right panel: the NBQX trace is almost completely hides the control trace. Perhaps use the same indicators [(1) control, (2) CTZ, (3) NBQX] as on the left panel.

Thanks, this is corrected.

Reviewer #2:In recent years the Trussell group has produced several interesting papers examining the mechanisms of synaptic transmission in MF-UBC synapses. In the present work, it is shown that under basal conditions, there is enough ambiant glutamate to significantly activate ionotropic and metabotropic receptors at these synapses. Furthermore, the authors convincingly show that the basal glutamate is provided by the continuous supply linked to spontaneous vesicular release from resting MF terminals. Finally, in a spectacular final figure, the authors present intriguing data suggesting that ambient glutamate may shift the phase of responsiveness of the UBC cells to periodic stimulation. This leads to the very interesting suggestion that the UBC to GC relay may shift the phase of the incoming cerebellar sensory input in a flexible manner.The design and execution of the experiments are excellent, and the conclusions of the paper are both novel and justified.My only substantial criticism concerns the treatment of glutamate diffusion in the glomerulus. In both Introduction and Discussion, the authors say that glutamate escape out of the glomerulus may occur slowly, on a time scale of 100s of ms. But their actual diffusion model, that is used in the Results section, does not incorporate a separate, slow phase representing glutamate exit out of the glomerulus. Rather, it treats glutamate diffusion as a single process that is homogeneous in space, with global slowing factors reflecting volume fraction and tortuosity. This results in an apparent contradiction that should somehow be addressed, at least with a caveat sentence when presenting the diffusion model, saying that this model is only a rough approximation.

We have now added caveats in the Materials and methods to acknowledge just this point.

I have listed below specific suggestions that should be taken into consideration when revising the manuscript.1) Glutamate diffusion model:Whereas the AMPAR activation is modelled in detail (Figure 2A), the diffusion of glutamate is described by a highly simplified model (diffusion equation). This model does not attempt to incorporate the complicated MF terminal geometry, which is understandable. However, it may be oversimplified, as all glutamate sources are considered equivalent, and the diffusion process is homogeneous in space.Incorporating crevices in the model, as was done by Nielsen et al. (2004), results in the prediction of two phases of glutamate concentration decay, with a slow phase reflecting glutamate escape once the glutamate concentration has equilibrated within the glomerulus. A biphasic glutamate decay model was likewise developed by Barbour et al. (1994) at PF-PC synapses. Such a biphasic decay could help interpreting the time course of mEPSCs. Specifically, it is possible that the slow time constant of mEPSC decay could reflect a slow component of glutamate concentration decay (escape out of the glomerulus), rather than a low value for channel resensitization rates, as it presently appears from Supplemental Table 1.

The situation is not so much “rather than” as “both”: The slow clearance of glutamate slows the EPSC and slows re-sensitization (since receptors cannot re-sensitize until transmitter levels fall). This is implicit in the model, and seems inescapable give the data shown here and in Lu et al., 2017.

The possibility of a slow glutamate escape would give a likely mechanistic interpretation for the finding of a significant basal glutamate concentration inside the synaptic cleft. By contrast, this conclusion of the paper is presumably difficult to reconcile with the monotonic glutamate decay assumed, given the scarcity of recorded mEPSCs.

The glutamate transient we assumed is monotonic in that it continuously decreases after the peak, but it does not decrease monophasically. The equations predict a significant slow tail. This slow phase is larger when there is multivesicular release within the “confines” of the MF-UBC synaptic cleft area, but even the recorded mEPSCs shows evidence of this slow phase.

Again, I do not suggest developing a realistic diffusion model based on glomerulus morphology: this would be a huge task that would require the introduction of many geometric parameters that are presently unknown. But the authors should acknowledge in their modelling section the possibility that glutamate concentration decay could be biphasic, and that this could be an explanation for the slow phase of mEPSC decay.

Again, the concentration decay is “biphasic” in the model. In the modeling figure panels E and I, for example, the predicted glutamate profile is plotted on a log scale and is clearly nonlinear on that axis. The slow “phase” lasts for hundreds of milliseconds. We have also explored models using multiexponential decays, which give very similar results. We have uploaded all of these models (kinetic, 3D and exponential) into Neuron Modeldb, and this link is cited in the paper. The conclusions are robust.

2) mGluR1 activation by basal glutamate?When reading the section on effects of basal glutamate in ON UBCs, the question arises as to whether mGluR1s would be activated in addition to AMPARs. Was this examined? Even if the results were negative, it would be good to mention them for completeness.

As requested, we have added such a sentence to the Results.

3) Figure 6:This figure has two parts: panels A-E describe mEPSCs in UBCs, and F-H describe combined bafilomycin/CTZ suggesting that the source of ambient glutamate is vesicular. It is the second part which is most directly relevant to the paper. While describing mEPSCs is valuable -and may yield important clues on glutamate diffusion, see above-, the fact that mEPSCs are present in these cells does not represent a surprise. In view of the above the authors should consider keeping only parts F-H as main text figure, and downgrading parts A-E as an additional supplementary figure.

We agree that these two parts of the figure describe somewhat different topics and so have split them into two figures. Regarding the “surprise” element, while we agree that mEPSCs are expected in most neurons, we were actually quite surprised to see them in UBCs: such events have not been reported in UBCs, and our initial efforts did not reveal any. Only when we realized how small and how incredibly slow they are (for mEPSCs) did we re-examine the datasets with greater filtering than typical for whole-cell recordings and then realized they were there all along. Since their presence is part of the logic of our proposal that spontaneous exocytosis supplies ambient glutamate, we prefer to let them stay in the paper.

4) Model of Figure 8:In the analysis of Figure 8E-F, the control response is modelled as a second harmonic of the MF spike modulation, while the response in TBOA is modelled as a first harmonic. But looking at the data in D-E, it appears that modelling the control response as a second harmonic is rather arbitrary. The amplitude of the second wave is much smaller than that of the first one, indicating that the true period remains the driving period. Looking at the raw data in D, it appears that the black spikes are roughly synchronized with the driving signal, except for a small delay, whereas the red spikes are almost opposite in phase with respect to the driving signal.

This is what we were are trying to convey.

This interpretation could be conveyed by performing a first harmonic fit to the black data as well as to the red data in E. Fitting the black histogram in E with a first harmonic would considerably simplify the presentation of the next panel (F).

We fit with a simple sine wave with a variable period. It is not clear how a first harmonic would be any different.

Presently panel F is almost impossible to understand. It is unclear what the radius in F represents (probably the firing rate for the second harmonic was doubled; but this is hard to justify). It is also unclear how the arc circles joining black and red dots were obtained. And it is unclear why the phase chosen for the black dots corresponds to the second peak in E rather than to the first one.

We have tried to explain this further in the legend and with explanatory changes in the figure panels E and F, emphasizing that we are plotting the phase of offset responses with and without TBOA and showing their accompanying firing rates.

Reviewer #3:This study shows how glutamate transporters limit activation of glutamate receptors at ON and OFF UBCs. The experiments show quite convincingly that glutamate transporters regulate ambient glutamate levels and synaptic glutamate diffusion dynamics. ON and OFF UBCs serve as controls for each other in various experiments, enabling the authors to distinguish effects on ambient and evoked transmitter levels. Further experiments show that ambient glutamate is due to spontaneous vesicle release and that transporters influence physiologically relevant UBC signaling. The experiments appear rigorously performed and are clearly presented. My primary concern is with the kinetic model used to estimate the glutamate concentration.The authors estimate the glutamate concentration time course using a kinetic model of AMPA receptors. The experimental bases for model parameters are not described thoroughly and raise many questions. The authors note that parameters were tuned to match whole-cell responses recorded from dissociated UBCs (Lu, et al., 2014), experiments in which glutamate was applied to the whole cell with relatively slow exchange times (estimated by the authors to be 50 ms). These methods are likely inadequate to constrain many of the parameters in the model. As just one example, the authors determine that the glutamate binding rate at room temperature to a single binding site is 2×108 s^-1^M^-1^s^-1^, which is certainly within theoretical diffusion limits but, as far as I know, is 10 times faster than any previous estimates and cannot be derived reliably from the cited whole-cell data. This is likely to be an important issue, because a higher binding rate would confer higher sensitivity in non-equilibrium conditions.

We repeated simulations and found that a 10-fold reduction in binding rate, accompanied by parallel reduction in dissociation, had almost no effect on the predicted responses. This is expected given that the “meat” of this work is in responses to slow glutamate transients. We note that association rates assumed for receptors are often in the 10^7^ / M.s range as the reviewer notes, but in many cases these are not actually measured. Measured values for ligand binding to ligand gated channels varies across the 10^7^-10^8^ range. Given all this, we feel comfortable that values utilized here do not weaken our conclusions.

I do not suggest that the authors undertake an entire kinetic analysis of UBC AMPA receptors, but I would ask that they add more detail supporting the rate constants and how much the model's behavior (beyond the non-monotonic dose-response) relies on specific values. I would also suggest that the model constitutes the weakest part of the paper and distracts somewhat from an otherwise pleasing flow of nice physiological experiments. I would suggest that the model is not necessary and, if retained, might function better as a discussion point at the end.

In response to the options offered by the editors we have moved the model to a supplemental figure. But to move it to a discussion point at the end of the paper would up-end the point of the model which was to make a testable prediction about ambient glutamate. Without that, the flow of the paper would consist of an investigation into a rather random query about the possibility of ambient glutamate. We note also that the model values chosen mimic extraordinarily diverse aspects of the receptors during fast and slow glutamate transients: the amplitudes of the fast and slow EPSCs and their time courses are almost perfectly fit by the model. The excellence of the fits across diverse time scales (ms to seconds), we believe, is supportive of the parameters of the kinetic model.